# De-repression of the RAC activator ELMO1 in cancer stem cells drives progression of TGFβ-deficient squamous cell carcinoma from transition zones

Heather A McCauley[1]*, Véronique Chevrier[2], Daniel Birnbaum[2], Géraldine Guasch[1,2]*

[1]Division of Developmental Biology, Cincinnati Children's Hospital Medical Center, Cincinnati, United States; [2]Centre de Recherche en Cancérologie de Marseille (CRCM), Inserm, U1068, F-13009, CNRS, UMR7258, F-13009, Institut Paoli-Calmettes, F-13009, Aix-Marseille University, UM 105, F-13284, Marseille, France

**Abstract** Squamous cell carcinomas occurring at transition zones are highly malignant tumors with poor prognosis. The identity of the cell population and the signaling pathways involved in the progression of transition zone squamous cell carcinoma are poorly understood, hence representing limited options for targeted therapies. Here, we identify a highly tumorigenic cancer stem cell population in a mouse model of transitional epithelial carcinoma and uncover a novel mechanism by which loss of TGF$\beta$ receptor II (*Tgfbr2*) mediates invasion and metastasis through de-repression of ELMO1, a RAC-activating guanine exchange factor, specifically in cancer stem cells of transition zone tumors. We identify ELMO1 as a novel target of TGF$\beta$ signaling and show that restoration of *Tgfbr2* results in a complete block of ELMO1 in vivo. Knocking down *Elmo1* impairs metastasis of carcinoma cells to the lung, thereby providing insights into the mechanisms of progression of *Tgfbr2*-deficient invasive transition zone squamous cell carcinoma.

*For correspondence: Heather. McCauley@cchmc.org (HAM); geraldine.guasch-grangeon@ inserm.fr (GG)

**Competing interests:** The authors declare that no competing interests exist.

## Introduction

Transition zones are distinct anatomical regions between two different types of epithelia, and function as stem cell niches in many regions of the body (*Runck et al., 2010*; *San Roman et al., 2011*; *Mcnairn and Guasch, 2011*). Populations of cells with stem cell characteristics, such as label-retention and self-renewal, and expression of stem cell markers such as the cell surface glycoprotein CD34, have been found in the transition zones between the cornea and conjunctiva in the eye (*Zieske, 1994*; *Cotsarelis et al., 1989*), between the esophagus and the stomach (*Kalabis et al., 2008*), between the endocervix and ectocervix (*Elson et al., 2000*; *Herfs et al., 2012*), between the mesothelium and oviductal epithelium of the ovary (*Flesken-Nikitin et al., 2013*) and between the anal canal and the rectum (*Runck et al., 2010*). Transition zones are uniquely susceptible to carcinogenesis, and the resulting tumors are typically highly malignant and associated with poor prognosis (*San Roman et al., 2011*; *Mcnairn and Guasch, 2011*; *Flesken-Nikitin et al., 2013*; *Grodsky, 1961*). Ocular surface squamous neoplasias are relatively rare, but most of them involve the limbus (*McKelvie et al., 2002*). As many as 86% of esophageal tumors arise in association with Barrett's esophagus at the esophageal-gastric junction (*Trudgill et al., 2003*; *Yamamoto et al., 2016*). The transition zone in the ovary was recently shown to be acutely sensitive to oncogenic transformation (*Flesken-Nikitin et al., 2013*). Cervical cancers arise at the transition between the columnar epithelium of the endocervix and the squamous epithelium of the ectocervix (*Elson et al., 2000*;

**eLife digest** Many different types of cells make up the tissues and organs throughout our bodies. There are locations throughout the body where two different types of cells meet – called transition zones – and these regions are susceptible to cancer formation. Many of these tumors are particularly aggressive, including those that arise in the transition zone in the cervix, the junction between the esophagus and the stomach, and the transition zone between the anus and rectum. Aggressive tumors such as these frequently spread and form tumors in other organs, such as the lung, in a process called metastasis.

We still lack a clear understanding of what makes transition zones prone to forming tumors or why the tumors that form are so aggressive. However, we do know about some differences between these tumor cells and healthy cells. For example, in healthy cells, the "transforming growth factor beta" (TGFβ) signaling pathway, is crucial for regulating many different processes, including cell growth. By contrast, many of the cells in aggressive transition zone tumors are unable to correctly regulate TGFβ signaling.

Mice that have been genetically engineered so that their cells are deficient in TGFβ signaling spontaneously develop aggressive transition zone tumors. By studying these mice, McCauley et al. found that only a small fraction of the tumor cells are responsible for the growth of the tumor. These cells express genes that enhance their ability to migrate and invade, including one called *Elmo1*. When McCauley et al. blocked the activity of this gene the aggressive tumor cells lost their ability to metastasize to the lung.

This is a new link in understanding how a particular genetic mutation or the inability to regulate a cell signaling pathway, such as TGFβ, can drive tumor growth and metastasis. Based on this knowledge, it may be worth investigating whether blocking the activity of ELMO1 could help to prevent metastasis from transition zone tumors.

*Herfs et al., 2012*; *Petignat and Roy, 2007*). Highly malignant squamous cell carcinomas (SCC) develop between the stratified epithelium of the anal canal and the simple epithelia of the rectum (*Grodsky, 1961*; *Kim et al., 2013*; *Guasch et al., 2007*).

To date, we still lack a clear understanding of the signaling and cellular mechanisms that drive transitional epithelial carcinogenesis. Deregulated TGFβ signaling seems to be a hallmark of aggressive transition zone cancers. In cervical cancer, mutations or loss of TGFβ downstream effectors SMAD2 and SMAD4 are common (*Maliekal et al., 2003*), and loss of nuclear SMAD2 and SMAD4 is associated with poor survival (*Kloth et al., 2008*). In genital SCC, TGFβ receptor (TGFβRII) protein expression is decreased and loss of phosphorylated SMAD2 is observed, even at early stages, suggesting that loss of TGFβ signaling may be an early event in carcinogenesis (*Guasch et al., 2007*). Mouse models targeting components of the TGFβ signaling pathway have been generated (*Muñoz et al., 2006*). Many epithelia develop normally despite the loss of a component of the TGFβ signaling pathway (*Guasch et al., 2007*; *Muñoz et al., 2006*; *Biswas et al., 2004*; *Padua and Massagué, 2009*). However, tumorigenesis occurs rapidly when these epithelia are exposed to carcinogens (*Biswas et al., 2004*), polyomavirus middle T antigen expression (*Forrester et al., 2005*), oncogenic mutations, such as mutations in *APC* (*Muñoz et al., 2006*), or activated *Hras* (*Guasch et al., 2007*; *Schober and Fuchs, 2011*; *Lu et al., 2006*) or *Kras* (*Lu et al., 2006*), or spontaneously in transition zones. Within the gastric transition zone, loss of SMAD3 (*Nam et al., 2012*) or BMP signaling (*Bleuming et al., 2007*) results in invasive carcinoma. Mice with a neuronal-specific deletion of *Tgfbr1* develop spontaneous periorbital and perianal SCC (*Honjo et al., 2007*). The backskin of mice lacking *Tgfbr2* in all Keratin 14-expressing (K14+) progenitors of the stratified epithelia is morphologically normal, but these mice develop spontaneous SCC in cervical and anorectal transition zones (*Guasch et al., 2007*).

RHO and RAC-guanine triphosphatases (GTPases) are small G proteins (21–25 kDa), and belong to the RAS superfamily (*Parri et al., 2010*). They act as molecular switches to elicit rapid changes in cell shape, polarity, and migratory ability in response to external cues (*Parri et al., 2010*; *Vega and Ridley, 2008*; *Sadok et al., 2014*; *Alan and Lundquist, 2013*) and are major players in malignant

cell invasion. RAC exists in an inactive form, bound to GDP, and in an active form, bound to GTP (*Parri et al., 2010*; *Sadok et al., 2014*; *Laurin and Cote, 2014*; *Lazer and Katzav, 2011*). Guanine exchange factors (GEFs) are required to promote the active, GTP-bound form of RAC, and GTPase activating proteins (GAPs) return RAC to its inactive, GDP-bound state (*Parri et al., 2010*; *Vega and Ridley, 2008*; *Sadok et al., 2014*; *Laurin and Cote, 2014*). More than 70 GEFs have been described, which act downstream of many signaling pathways, including growth factor receptors, integrins, cadherins, and cytokine receptors (*Parri et al., 2010*). Engulfment and cell motility (ELMO) proteins (originally described as CED-12 in *C. elegans)* participate in RAC1-dependent engulfment and apoptosis (*Côté and Vuori, 2007*; *Gumienny et al., 2001*). ELMO proteins form a complex with DOCK proteins that serves as a GEF for RAC proteins. This complex plays important roles in chemotaxis, phagocytosis, neurite outgrowth, and cancer cell invasion (*Laurin and Cote, 2014*; *Côté and Vuori, 2007*; *Gumienny et al., 2001*; *Grimsley et al., 2004*; *Brugnera et al., 2002*; *Jarzynka et al., 2007*; *Sai et al., 2008*; *Li et al., 1706*; *Komander et al., 2008*).

Subsets of long-lived tumor-initiating stem cells or cancer stem cells (CSCs) are often resistant to cancer therapies and thus may be responsible for tumor recurrence (*Clevers, 2011*; *Malanchi et al., 2012*). They sustain tumor growth through their ability to self-renew and to generate differentiated progeny, and they may play a role in metastasis (*Clevers, 2011*; *Malanchi et al., 2012*; *Oskarsson et al., 2014*; *Chaffer and Weinberg, 2011*; *Charafe-Jauffret et al., 2010*). To date, the cellular and molecular mechanisms of *Tgfbr2*-deficient transition zone carcinoma development and metastasis are unknown. In this study, we used the murine *Tgfbr2*-deficient anorectal SCC model to study the consequences of loss of TGFβ signaling in CSC-driven tumor propagation and metastasis. We found that these *Tgfbr2* cKO anorectal SCC, which spontaneously metastasize to the lungs, contain a unique population of epithelial cells with features of CSCs, including: expression of the CSC marker CD34, clonogenicity in vitro, tumorigenicity in vivo, and upregulation of genes associated with invasion and metastasis. Using RNA-Sequencing and chromatin immunoprecipitation, we uncovered a novel mechanism linking loss of TGFβ signaling with invasion and metastasis via the RAC-activating GEF ELMO1. We show that *Elmo1* is a novel target of TGFβ signaling via SMAD3 and that restoration of *Tgfbr2* results in complete block of ELMO1 in vivo. Knocking down *Elmo1* impairs metastasis to the lung, providing a new therapeutic avenue to target the early phase of metastasis in highly aggressive transition zone tumorigenesis.

## Results

### *Tgfbr2*-deficient anorectal SCC contain a distinct population of cells with clonogenic potential

Mice lacking *Tgfbr2* in stratified epithelia expressing Keratin 14 (K14) develop spontaneous squamous cell carcinoma (SCC) at the transition zone between the anal canal and rectum (*Guasch et al., 2007*). To lineage trace *Tgfbr2*-deficient cells within these transitional SCC and enable isolation of specific *Tgfbr2*-deficient tumor cell populations, we backcrossed these mice into mice containing loxP sites flanking a STOP sequence preceding eYFP inserted into the *Rosa26* locus (*Figure 1—figure supplement 1*), such that all K14-positive epithelial cells, including the anorectal SCC cells, while conditionally null for *Tgfbr2* expressed YFP (cKO mice, *Figure 1A–C*). We had previously identified a population of cells with stem cell characteristics, including colocalization with known stem cell markers, such as CD34, in the anorectal transition zone of wild-type mice (*Runck et al., 2010*). We hypothesized that tumors arising at the anorectal transition zone in the *Tgfbr2* cKO mice would contain a population of CD34-expressing cells, and that these cells would represent a population of tumor-propagating cells or so-called cancer stem cells (CSCs). Based on the idea that CSCs should reside at the tumor–stroma border, we thought that CSCs of anorectal SCCs should express abundant integrins. To test this hypothesis, we first analyzed marker expression within histologic sections of *Tgfbr2*-deficient anorectal tumors. Immunofluorescence staining revealed that all YFP⁺SCC cells located at the tumor–stroma interface expressed high levels of the hemidesmosomal α6 integrin (*Figure 1D*) and the focal adhesion marker β1 integrin (*Figure 1E*), and a fraction of these expressed CD34 (*Figure 1F*). These three markers were used to isolate discrete populations of cells by fluorescence-activated cell sorting (FACS). Tumors were dissociated into a single-cell suspension as previously described (*McCauley and Guasch, 2013*), stained with antibodies, and subjected to flow

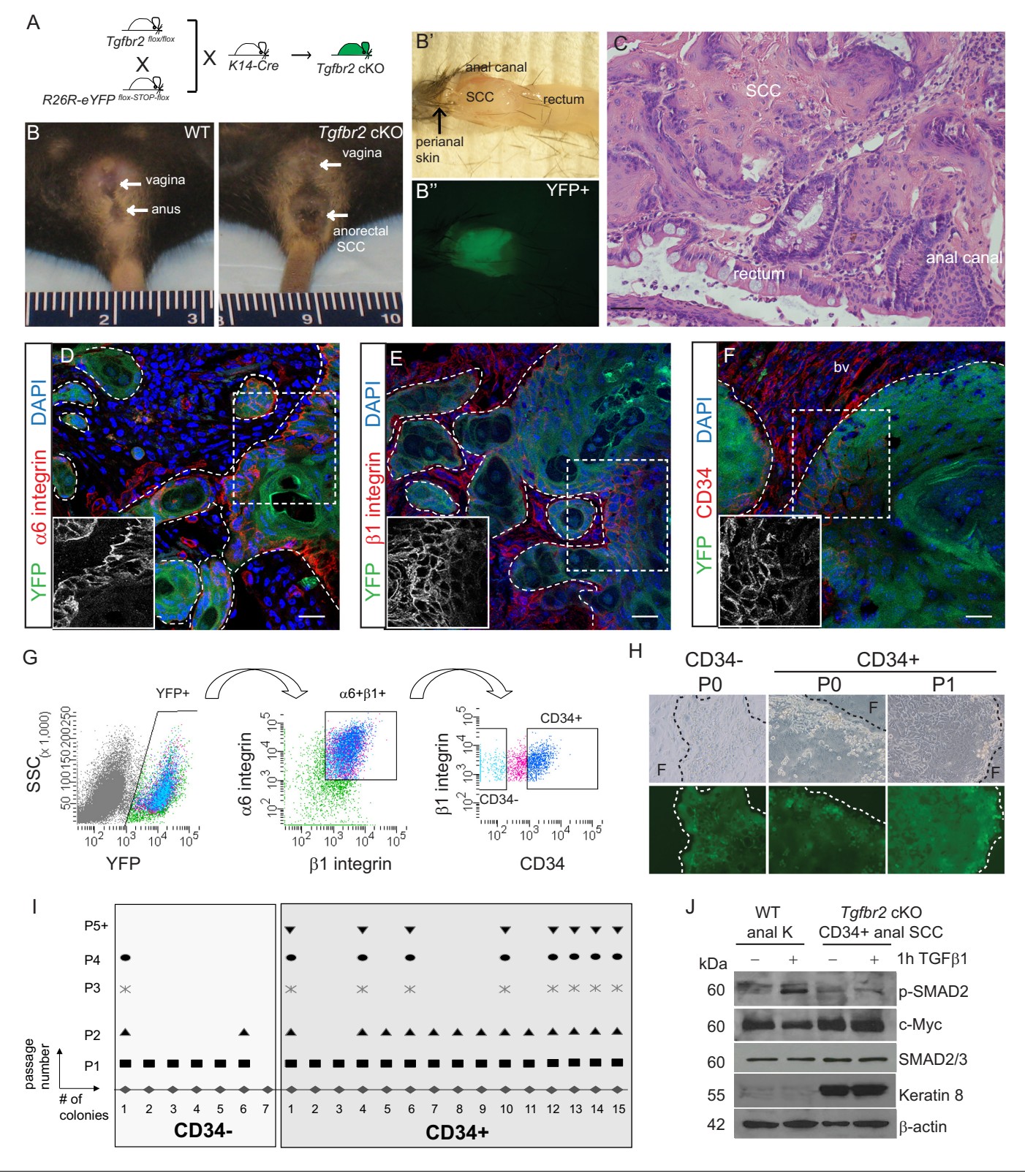

**Figure 1.** *Tgfbr2* cKO anorectal SCC contain a population of epithelial CD34+ cells which are enriched for in vitro clonogenicity. (**A**) Triple transgenic mice were obtained by crossing *Tgfbr2flox/flox* mice with *R26R-eYFPflox-STOP-flox* mice and *K14-Cre* mice. (See ***Figure 1—figure supplement 1***) (**B**, **B'**, **B''**). *Tgfbr2* cKO mice developed spontaneous anorectal tumors (**B**) which formed in the anal canal between the hair-bearing perianal skin and rectum (**B'**), and expressed YFP (**B''**). (**C**) Hematoxylin and eosin (H and E) staining of *Tgfbr2* cKO tumors revealed moderately- to poorly-differentiated
*Figure 1 continued on next page*

*Figure 1 continued*

squamous cell carcinoma (SCC) at the transition between the epithelium of the anal canal and the rectum. (D–F) Immunofluorescence staining of *Tgfbr2* cKO anorectal SCC revealed YFP+ tumor cells expressed both α6-integrin (D) and β1-integrin (E) and that there was a distinct population of CD34+ tumor cells (F). Boxed areas represent isolation and magnification of the red channel. DAPI counterstains nuclei in blue. Representative of 16 mice analyzed by histology and immunostaining and 37 analyzed by FACS (G). CD34+ and CD34− cells were isolated from the *Tgfbr2* cKO anorectal SCC by fluorescence-activated cell sorting (FACS). After dissociation and staining, CD45+ blood cells, CD31+ endothelial cells and CD11b+ macrophages were excluded from the live (7AAD-), K14+YFP+ population. These epithelial tumor cells were further purified by gating for α6-integrin (CD49f+) and β1-integrin (CD29+) cells (YFP +7AAD-CD11b-CD31-CD45-CD49f+CD29+, abbreviated: YFP+). Of these epithelial tumor cells, distinct CD34+ and CD34− populations were isolated (see *Figure 2—figure supplement 1*). (H–I) When plated on a feeder layer of irradiated fibroblasts, sorted YFP+CD34− and YFP+CD34+ tumor cells were able to form clones. CD34− clones appeared differentiated and were unable to survive multiple passages, whereas CD34+ clones formed holoclones and subsequently robust cell lines which survived unlimited passage (n = 4 tumor-bearing mice). Panel H is a representative example of three different primary clones isolated from one of four distinct spontaneous tumors. (J) *Tgfbr2* cKO anorectal CD34+ SCC cells are nonresponsive to TGFβ as they did not phosphorylate SMAD2 nor downregulate c-Myc compared to WT anal keratinocytes, and aberrantly expressed Keratin 8. Abbreviations: bv, blood vessel; F, irradiated fibroblast; P0, passage 0; P1, passage 1; k, keratinocyte. Scale bars = 20 µm.

The following figure supplement is available for figure 1:

**Figure supplement 1.** Conditional targeting of *Tgfbr2* and lineage tracing in Keratin 14-positive tissues.

cytometry. Blood cells, endothelial cells, macrophages and dead cells were excluded, and live, K14 +YFP+ epithelial cells were further purified upon α6-integrin and β1-integrin expression. Of these live, YFP+, α6-integrin+, β1-integrin+ cells, distinct CD34-negative (CD34−) and CD34-positive (CD34+) populations of cells were observed and isolated (*Figure 1G*). The frequency of epithelial CD34+ cells within the tumor varied between mice, from 7% to 34%. When plated on a feeder layer of irradiated fibroblasts, both YFP+CD34−and YFP+CD34+ cells sorted from *Tgfbr2* cKO SCC formed colonies; however, CD34− colonies appeared to be differentiated paraclones and were unable to be passed more than once, whereas CD34+ colonies appeared to form holoclones, were able to proliferate extensively, and survived unlimited passage (*Figure 1H–I*). CD34+ SCC cells did not respond to TGFβ stimulation, confirming the loss of TGFβRII, and aberrantly expressed Keratin 8, a hallmark of SCC (*Ikeda et al., 2008*), compared to keratinocytes isolated from the anal canal of wild-type mice (*Figure 1J*). These data suggest that the CD34+ population of epithelial cells isolated from the *Tgfbr2* cKO anorectal SCC has self-renewal properties in vitro.

## *Tgfbr2* cKO CD34+ SCC cells are enriched for in vivo tumorigenicity

We next sought to determine whether the *Tgfbr2*-deficient CD34+ SCC cells were enriched for self-renewal properties and tumorigenicity in vivo. We previously developed an orthotopic transplantation assay in which anorectal SCC cells are injected specifically and reproducibly within the anorectal transition zone of immunocompromised *Nu/Nu* mice (*McCauley and Guasch, 2013*). Orthotopic injection of 200–$10^6$ CD34+ cells from *Tgfbr2* cKO anorectal SCC into recipient *Nu/Nu* mice caused secondary tumor formation with 100% efficiency (n = 89) (*Figure 2—figure supplement 1A*). These secondary anorectal tumors were invasive, moderately- to poorly-differentiated SCC, characterized by cellular atypia, squamous nests with keratin pearls, intercellular bridges, aberrant mitoses, cellular disorganization and desmoplastic stroma (*Figure 2—figure supplement 1C*), and were morphologically similar to the primary *Tgfbr2* cKO tumors of origin. Just as in the primary *Tgfbr2* cKO SCC, a population of YFP+ epithelial cells expressing CD34 could be identified by immunofluorescence staining (*Figure 2B*) and isolated by FACS using the same strategy as used previously (*Figure 2C*). The frequency of epithelial CD34+ cells in the secondary tumors ranged from 7% to 22%. CD34 expression within this population of cells was confirmed at the mRNA level (*Figure 2—figure supplement 2*). We confirmed that loss of *Tgfbr2* was maintained in YFP+ SCC cells by qPCR (*Figure 2D*) and immunofluorescence staining, whereas TGFβRII and nuclear pSMAD2 were still present in the surrounding stroma (*Figure 2E–F*).

To determine whether SCC CD34+ cells were tumorigenic, CD34+ and CD34− YFP+ epithelial cells were isolated from the secondary anorectal tumor by FACS and orthotopically transplanted into the anorectal transition zone of *Nu/Nu* mice. After tertiary transplant of CD34+ SCC cells, 8/13 mice developed anorectal tumors, compared to 1 mouse of 13 transplanted with CD34− SCC cells

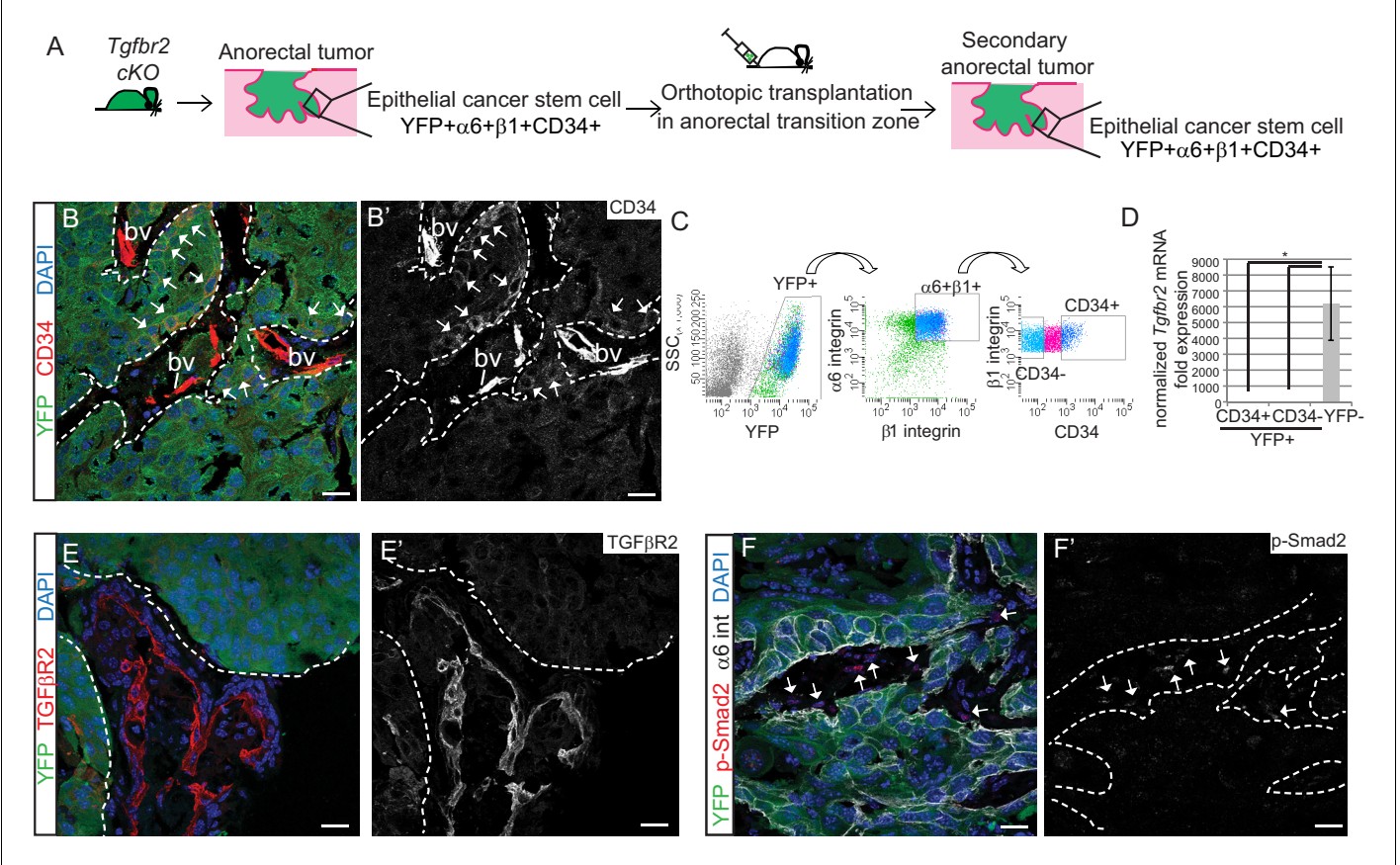

**Figure 2.** *Tgfbr2* cKO CD34+ SCC cells are enriched for in vivo tumorigenicity. (**A**) Strategy to generate secondary tumors from the triple transgenic mice *K14-Cre; Tgfbr2flox/flox; R26R-eYFPflox-STOP-flox* (cKO). (**B–B'**) Immunofluorescence staining of the secondary anorectal SCC revealed populations of YFP+CD34+ and YFP+CD34− tumor cells, preserving the hierarchy observed in the primary *Tgfbr2* cKO tumor. White arrows show the clusters of YFP+CD34+ cells. Dotted lines delineate the tumor from stroma. DAPI counterstains nuclei in blue. See *Figure 2—figure supplement 2* for histology and FACS profile of the secondary and tertiary tumors. (**C**) Using the same FACS strategy as employed for the primary *Tgfbr2* cKO anorectal SCC, the secondary anorectal tumors were sorted and distinct CD34+ and CD34− epithelial populations were isolated. (**D**) FACS-isolated YFP+CD34+ and YFP+CD34− epithelial tumor cells were subjected to mRNA extraction and qPCR and compared to FACS-isolated YFP-negative cells for *Tgfbr2* expression. Data represent the mean ± s.d. from three independent tumors; Student's *t*-test, *p=0.0313. (**E–F**) Immunofluorescence staining of the secondary anorectal SCC confirmed the loss of TGFβRII (**E–E'**) and phosphorylated SMAD2 (**F–F'**) in the epithelial YFP+ cells while expression was maintained in the K14-YFP- stroma (denoted by the white arrows). This is a representative example of 21 secondary tumors analyzed by histology, immunostaining and FACS. Abbreviation: bv, blood vessel. Scale bars = 20 μm.

The following source data and figure supplements are available for figure 2:

**Source data 1.** Values and statistics for *Figure 2D* using the Wilcoxon matched-pairs signed rank test.

**Figure supplement 1.** Orthotopic transplant of CD34+ cKO SCC cells results in secondary and tertiary tumor formation which recapitulate the hierarchy of the tumor of origin.

**Figure supplement 2.** *CD34* mRNA expression correlates with CD34 protein expression in sorted cKO SCC cells.

(*Figure 2—figure supplement 1B*). YFP+ cells negative for α6-integrin and β1-integrin were unable to form tumors upon transplantation (n = 14), indicating that not every cell type within the *Tgfbr2* cKO anorectal SCC is tumorigenic. Tertiary anorectal tumors maintained the tumor hierarchy of the primary and secondary *Tgfbr2* cKO SCC, and selection for CD34+ CSCs resulted in an increased ratio of CD34+ cells (75%) (*Figure 2—figure supplement 1D–E*). In fact, in the single tertiary tumor that formed after transplant of CD34− SCC cells, CD34 was re-expressed in 49% of YFP+ epithelial

tumor cells by FACS (*Figure 2—figure supplement 1G*), indicating that CD34 expression is dynamic in vivo. This is in accordance with the dynamic CD34 expression found in a DMBA-induced model of *Tgfbr2* deficient SCC of the backskin (*Schober and Fuchs, 2011*). Whereas CSC marker expression may be dynamic (*Clevers, 2011*), CD34 remains a useful marker to assay the CSC properties of a population of cells isolated from murine SCC. Taken together, the ability of CD34+ cells, which lack TGFβ signaling, to form secondary and tertiary tumors that recapitulate the hierarchy of the *Tgfbr2* cKO primary tumors suggests that CD34+ SCC cells are able to self-renew and differentiate in vivo.

## Transcriptional profiling of anorectal CSCs identifies a metastatic signature and upregulation of RAC signaling at the invading front of *Tgfbr2* cKO SCC

Squamous cell carcinomas, including those occurring at transition zones, frequently metastasize to the lung. We analyzed the lungs of tumor-bearing mice and observed that *Tgfbr2* cKO SCC indeed spontaneously metastasized to the lungs with 100% frequency (n > 30 mice analyzed) (*Figure 3A*). These lung metastases expressed Keratin 5 (*Figure 3—figure supplement 1*), indicating their squamous epithelial origin. Furthermore, *Tgfbr2* cKO lung metastases expressed YFP and contained populations of CD34+ cells (*Figure 3B–C*), recapitulating the hierarchy of the primary tumor of origin. Sequencing of RNA from CD34+ and CD34- *Tgfbr2* cKO SCC cells, isolated by FACS as described in *Figure 2C*, revealed that CD34+ SCC cells were enriched for an invasive and metastatic gene signature as well as for mRNA involved in the RAC/RHO/RAS pathway (*Figure 3D* and *Supplementary file 1*). Using ToppCluster to generate a network of genes shared between *Tgfbr2* deficient CD34+ anorectal transitional SCC cells and published datasets of aggressive human cancers (*Figure 3—figure supplement 2*) we found that a number of genes were commonly overexpressed in human cancers including cervical carcinoma. We validated a selection of the most upregulated genes in the CD34+ SCC cells by qPCR (*Figure 3E*), including *Cathepsin S, Fibrillin1, Spp1, Mmp9* and *Tgfb2*, which are all implicated in ECM organization, invasion and metastasis, and members of the RHO GTPase pathway *Rac2, Rhoh, RhoJ, Vav1, Dock2,* and *Elmo1*.

We confirmed that the GEF ELMO1 is co-expressed with CD34-positive tumor epithelial cells at the protein level by immunofluorescence staining (*Figure 4A*) and observed increased staining of this RAC-activating factor at the leading edge of the tumor. We also confirmed that RAC2 is co-expressed with CD34-positive cells (*Figure 4B*) and that RAC1 is strongly expressed at the tumor-stroma border of *Tgfbr2* cKO SCC (*Figure 4C*). A strong RAC activity was confirmed in vitro when we analyzed the amount of GTP-bound RAC in *Tgfbr2* cKO SCC CD34+ cells (*Figure 4D*). These results indicate that, while *Rac1* mRNA was not found to be upregulated in CD34+ cells by RNA-Seq analysis, RAC1 protein activity may be elevated in CSCs by upregulation of GEFs. Taken together, these data implicate CD34+ CSCs in metastasis of *Tgfbr2*-deficient SCC, potentially through upregulation of the RHO/RAC GTPase pathway.

## Upregulation of RAC signaling is unique to the anorectal *Tgfbr2* cKO SCC and not found in *Hras*-induced skin SCC

We hypothesized that aberrant RAC signaling was, at least in part, responsible for the highly aggressive nature of transition zone cancers. We compared the profile of CD34+ CSCs from our *Tgfbr2* cKO anorectal SCC to the CSCs purified from malignant skin SCC in mice from various genetic backgrounds treated with the chemical mutagen 7,12-dimethyl-benz[a] anthracene (DMBA) to induce an activating mutation in *Hras* (*Schober and Fuchs, 2011*). In DMBA-induced *Tgfbr2* cKO skin SCC, Schober et al reported two CSC populations: CD34^Hi and CD34^Lo, with the CD34^Lo CSC population demonstrating increased tumorigenicity over the CD34^Hi population. Bioinformatics analysis revealed that only six genes were common between the CSC from *Tgfbr2* cKO anorectal CD34+ cells and DMBA-induced *Tgfbr2* cKO skin CD34^Hi population (***p value=$1.57 \times 10^{-7}$) and seven genes were common in the DMBA-induced *Tgfbr2* cKO skin CD34^Lo population (***p value=$7.48 \times 10^{-5}$) (*Figure 5—figure supplement 1*). As expected in malignant cancers, these genes are involved in ECM organization, epithelial to mesenchymal transition (EMT) and metastasis but none was related to the RAC/RHO/RAS pathway. When we compared the skin CSC population from DMBA-induced *Tgfbr2* and focal adhesion kinase (FAK) double KO mice, in which SCC initiation is delayed compared to *Tgfbr2* cKO skin, nine genes were common between the *Tgfbr2* cKO anorectal

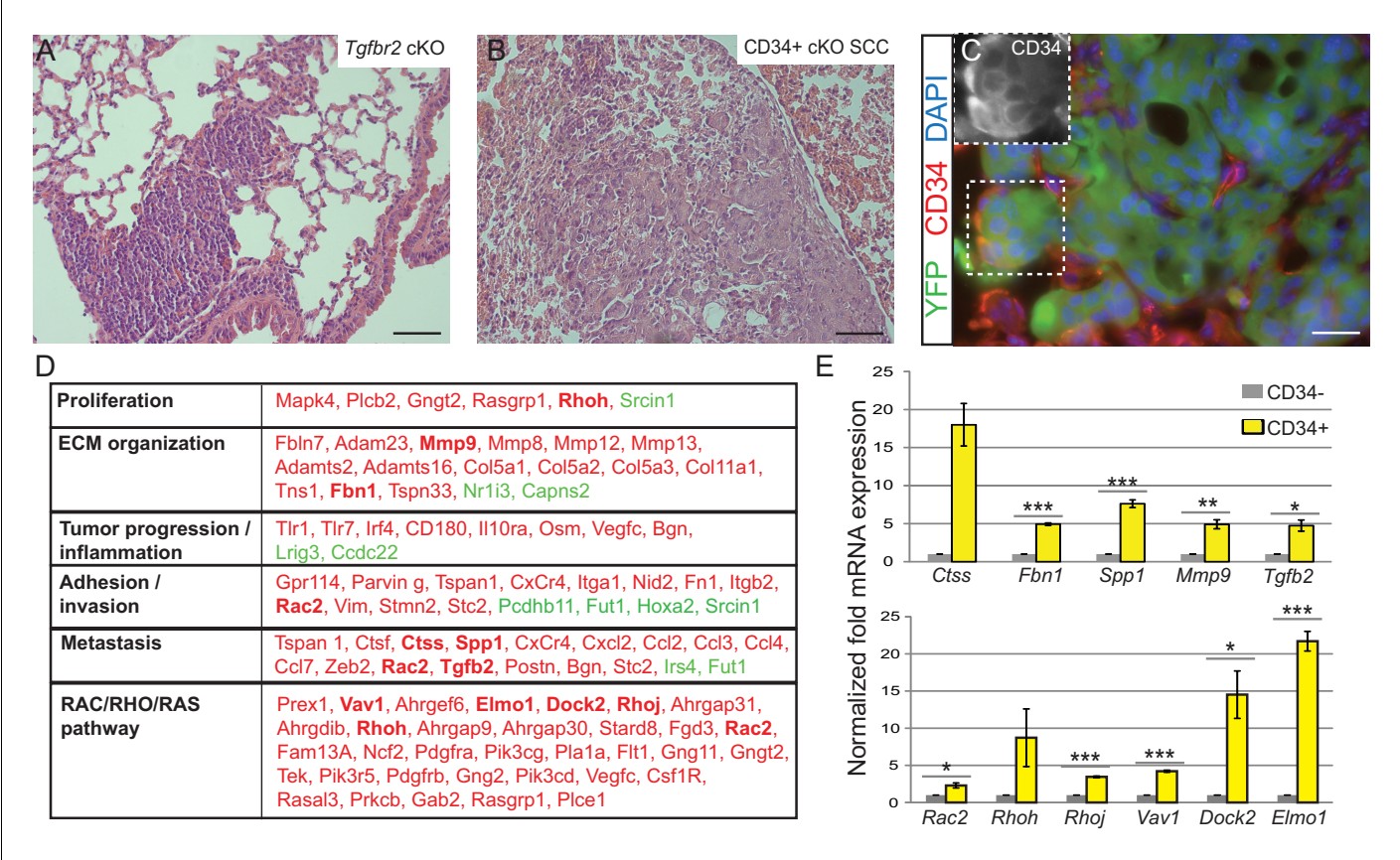

**Figure 3.** *Tgfbr2* cKO CD34+ SCC cells display a metastatic transcriptional signature. (A–C) H and E staining of the lungs of *Tgfbr2* cKO mice (A) and mice orthotopically transplanted with *Tgfbr2* cKO CD34+ SCC cells (B) revealed metastatic nodules which are YFP+ and contain a population of CD34+ tumor cells (C). The boxed area represents isolation and magnification of CD34 in the red channel. See also *Figure 3—figure supplement 1*. (D) RNA-Seq comparison of CD34− and CD34+ cells isolated from *Tgfbr2* cKO CD34+ SCC (n = 2 tumors each from two distinct cell lines) revealed that CD34+ SCC cells were enriched for an invasive and metastatic signature. This table represents a selected set of genes which are upregulated (red) or downregulated (green) by more than two fold with an FDR <0.05 in FACS-purified CD34+ cells compared to CD34− cells. Genes in bold were selected for validation by qRT-PCR. See *Supplementary file 1* for the full table of differentially expressed genes and *Figure 3—figure supplement 2* for comparison with human databases. (E) Selected genes which were upregulated in CD34+ cells compared to CD34− cells in the RNA-Seq analysis were selected for validation by qRT-PCR, including genes involved in ECM organization, adhesion, invasion and metastasis and the RAC/RHO/RAS pathway. Asterisks denote statistical significance using two-tailed, unpaired student's *t*-test; *Ctss* p=0.050895, *Fbn1* p***=0.00003, *Spp1* p***=0.00035, *MMP9* p**=0.00801, *Tgfb2* p*=0.016721, *Rac2* p*=0.0296, *Rhoh* p=0.177, *Rhoj* p***=0.000057, *Vav1* p***=0.000032, *Dock2* p*=0.02782, *Elmo1* p***=0.00067.

The following figure supplements are available for figure 3:

**Figure supplement 1.** Lung metastases express keratin 5.

**Figure supplement 2.** *Tgfbr2* cKO CD34+ SCC cells upregulate genes implicated in invasive human cancers.

CD34+ CSCs and the CD34$^{Lo}$population (***p value=1.98×10$^{-10}$). Similarly, only 11 genes involved in ECM organization and other functions were common between the *Tgfbr2* cKO anorectal CD34+ CSCs and the DMBA-induced skin CD34$^{Lo}$population in genetically wild-type mice (***p value=1.49×10$^{-11}$). The analysis of skin SCC signatures from FAK single KO mice, which are more refractory to DMBA-induced SCC formation, showed 21 common genes between *Tgfbr2* cKO anorectal CD34+ CSCs and the CD34$^{Lo}$population (***p value=5.35×10$^{-26}$) and one gene related to the RAC/RHO/RAS pathway was found in common (*Pdgfrb*). We also compared our *Tgfbr2* cKO anorectal SCC CD34+ CSC gene signature with DMBA-induced skin SCC CD34+ cells with overexpression of VEGF, which accelerates tumor growth (*Beck et al., 2011*). We found 40 genes that are upregulated in *Tgfbr2*-deficient anorectal CD34+ CSCs yet downregulated by VEGF in comparison to

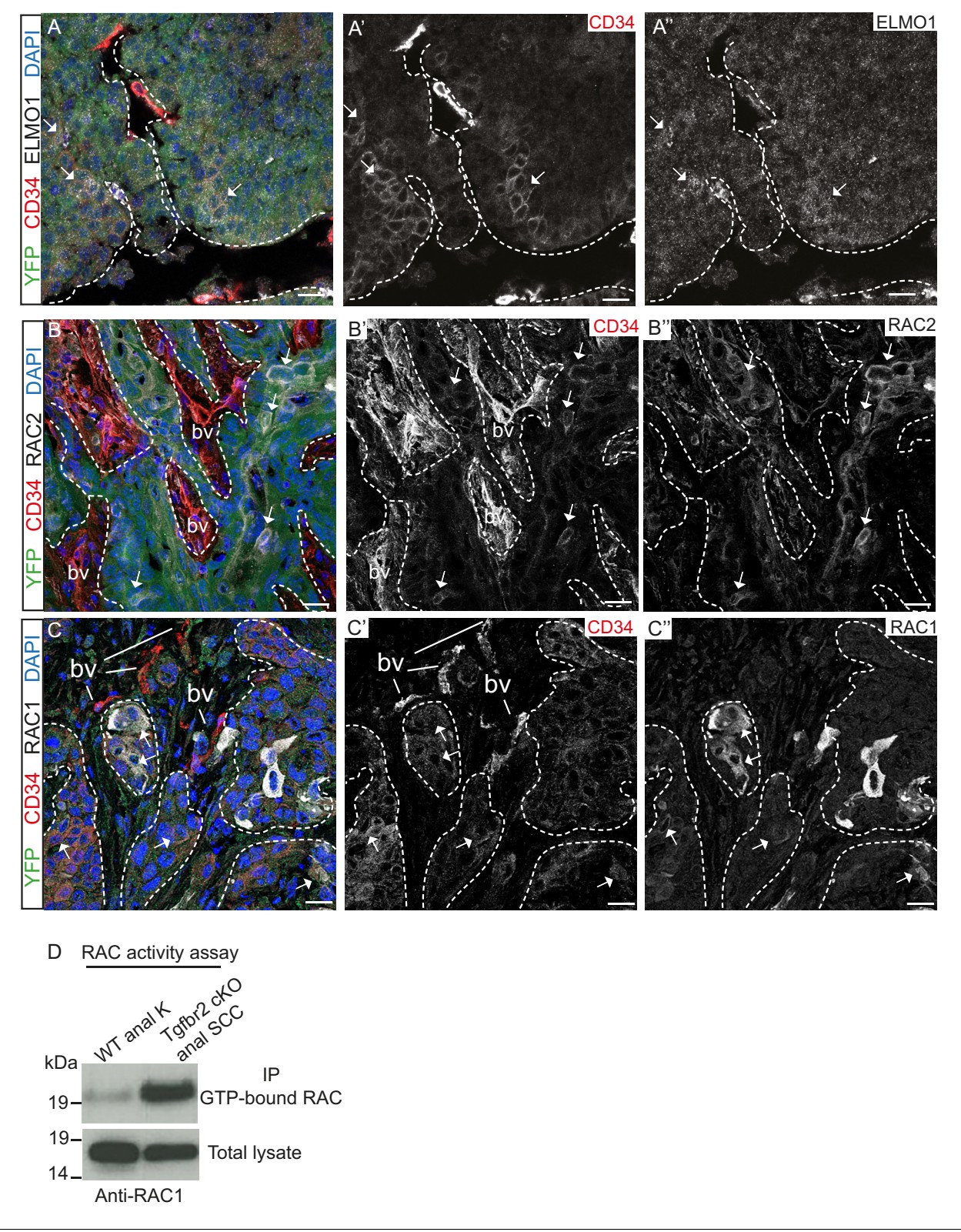

**Figure 4.** *Tgfbr2*-deficient tumors upregulate RAC signaling. (A–C) Immunofluorescence staining with antibodies against YFP, ELMO1, RAC2, RAC1 and CD34 revealed that some CD34+ tumor cells co-express ELMO1 (A, white arrows), RAC2 (B, white arrows) and RAC1 (C, white arrows), with strong expression at the invading front at the tumor-stroma border (dashed lines). DAPI counterstains nuclei in blue. n = 6 tumors tested. Abbreviations: bv,

*Figure 4 continued on next page*

Figure 4 continued

blood vessel. Scale bars = 20 µm. (D) RAC activity assay revealed the increased amount of GTP-bound RAC in the *Tgfbr2* cKO SCC CD34+ cell line compared to wild-type anal keratinocytes (K).

DMBA-induced backskin CD34+ CSCs (***p value=$2.69 \times 10^{-53}$), and we found 45 genes commonly upregulated between *Tgfbr2*-deficient anorectal CD34+ CSCs and DMBA-induced SCC CD34+ skin CSCs when mice overexpressed VEGF (***p value=$2.62 \times 10^{-39}$). Among these data sets, we found overlap of three RAC/RHO/RAS family genes with *Tgfbr2*-deficient anorectal CD34+ CSCs, suggesting that dysregulation of the RAC/RHO/RAS pathway may be associated with CSCs within highly aggressive tumors.

Immunofluorescence staining confirmed that TGFβ-deficient DMBA-induced skin SCC does not express RAC1 and RAC2 in contrast to anorectal SCC, where these proteins are found at the invasive front of the tumor (*Figure 5*). These analyses suggest that aberrant RAC signaling may be a hallmark of highly aggressive tumors arising spontaneously in tumor-prone transition zones.

## The GEF ELMO1 is a novel target of TGFβ signaling via SMAD3

ELMO proteins form a complex with DOCK proteins that serves as a GEF for RAC proteins in many cellular processes, including cancer cell invasion (*Laurin and Cote, 2014*; *Côté and Vuori, 2007*; *Gumienny et al., 2001*; *Grimsley et al., 2004*; *Brugnera et al., 2002*; *Jarzynka et al., 2007*; *Sai et al., 2008*; *Li et al., 1706*; *Komander et al., 2008*). Because *Dock2* and *Elmo1* mRNA were both upregulated in *Tgfbr2* cKO CD34+ SCC cells, we wondered whether the loss of TGFβ signaling was responsible for the upregulation of these RAC pathway activators. We probed the promoter regions of the mouse *Dock2* and *Elmo1* genes for the consensus SMAD-binding element (SBE) GTCT (*Derynck et al., 1998*; *Massague and Wotton, 2000*) and the consensus TGFβ-inhibitory element (TIE) GNNTTGGNGN (*Kerr et al., 1990*; *Frederick et al., 2004*; *McCauley et al., 2014*). The *Dock2* promoter contained one TIE located 266 bp upstream of the transcriptional start site (TSS) and four SBEs located 207, 497, 559 and 709 bp upstream of the TSS (*Figure 6—figure supplement 1A*), and the *Elmo1* promoter contained one TIE located 902 bp upstream of the TSS and one SBE located 153 bp upstream of the TSS (*Figure 6A*). These sites were evolutionarily conserved and not found in repeat regions of the genome. Moreover, these sites were not found in the promoter of *Elmo2*, despite its high degree of similarity with *Elmo1*. We used chromatin immunoprecipitation (ChIP) to determine whether SMAD3, a canonical effector of TGFβ signaling, bound any of these potential SBEs or TIEs. SMAD3 bound to the TIE on the *Elmo1* promoter, but not to the SBE (*Figure 6B*), and did not bind to any of the sites identified on the *Dock2* promoter (*Figure 6—figure supplement 1A*). These results indicated that *Elmo1* is a previously unidentified direct target of TGFβ signaling via SMAD3.

## Restoration of TGFβRII results in complete block of ELMO1 in vivo

Because we identified a direct link between SMAD3 and the promoter region of *Elmo1*, we hypothesized that rescue of TGFβRII in *Tgfbr2*-deficient cells would reduce ELMO1 expression. We cloned the full-length *Tgfbr2* gene into the pLVX-IRES-mCherry lentiviral vector and infected *Tgfbr2* cKO CD34+ SCC cells with this construct or the empty vector. Infection of YFP+CD34+ SCC cells with the rescue construct, but not the empty vector, efficiently restored *Tgfbr2* mRNA expression (*Figure 6C–D*). Rescued CD34+ SCC cells became sensitive to TGFβ treatment and phosphorylated SMAD2 (*Figure 6E*). Orthotopic transplantation of rescued CD34+ SCC cells resulted in a two-fold delay in tumor latency compared to *Tgfbr2* cKO CD34+ SCC cells infected with the empty vector (*Figure 6—figure supplement 2*), although all mice eventually developed tumors due to the inefficient infection rate of the rescue vector (*Figure 6F*). Tumors generated from the orthotopic transplantation of *Tgfbr2* cKO CD34+ SCC cells infected with the empty vector or rescued with full-length *Tgfbr2* were dissociated and YFP+mCherry+, YFP+mCherry-CD34+ and YFP+mCherry-CD34− cells were isolated by FACS (*Figure 6F*) and subjected to RNA extraction. YFP+mCherry+ cells isolated from tumors generated from rescued *Tgfbr2* cKO CD34+ SCC cells infected with full-length *Tgfbr2* expressed *Tgfbr2* mRNA 250-fold over YFP+mCherry- cells isolated from the same

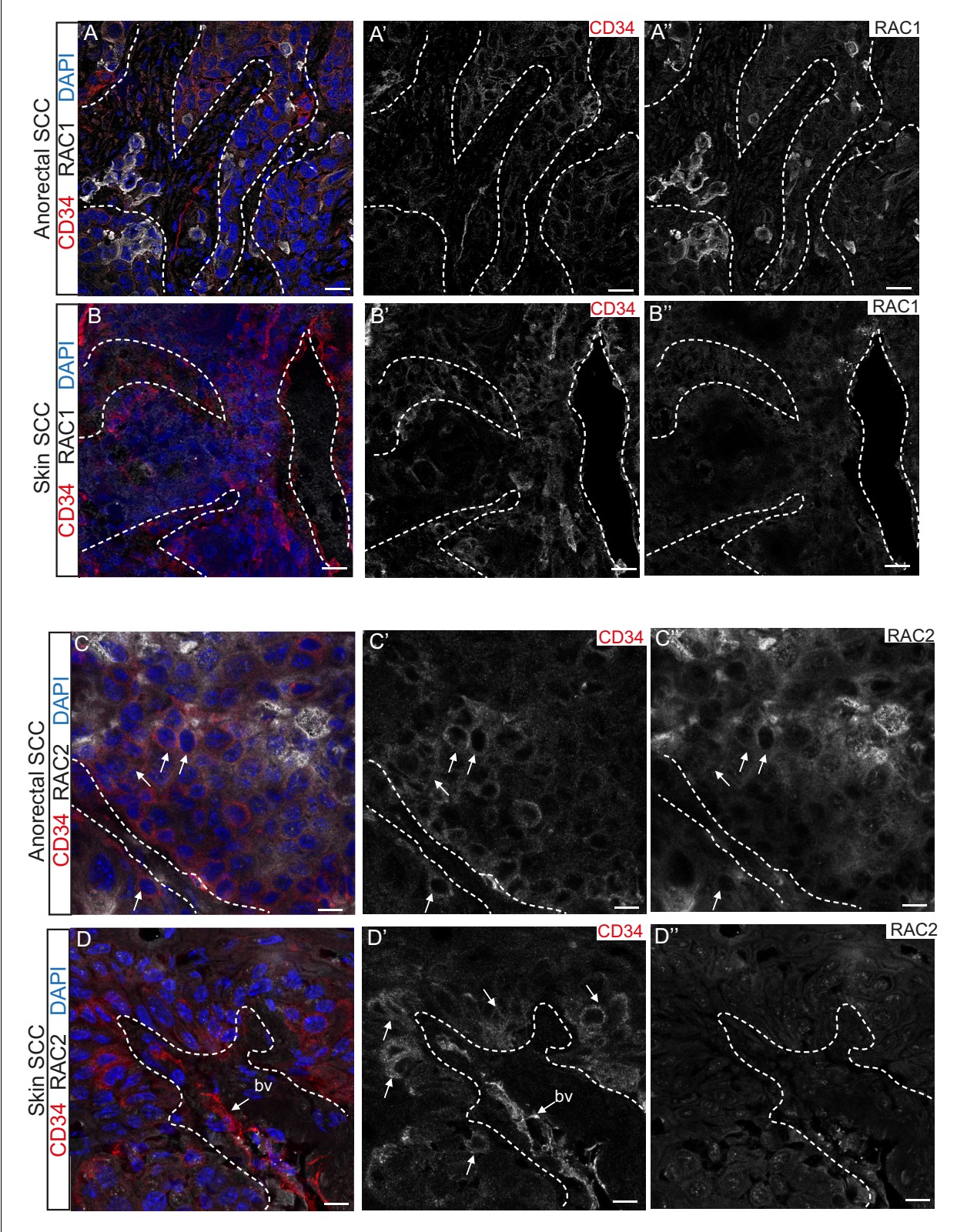

**Figure 5.** RAC1 and RAC2 are uniquely expressed in *Tgfbr2*-deficient transition zone tumors compared to DMBA-induced *Tgfbr2*-deficient skin SCC. (A–D) Immunofluorescence staining with antibodies against RAC1, RAC2 and CD34 revealed strong expression of RAC1 and RAC2 in the anorectal SCC tumor compared to the backskin SCC. Skin tumors come from cKO mice treated for 16 weeks topically with the chemical mutagen 7,12-dimethyl-benz [a] anthracene (DMBA) as previously described (*Guasch et al., 2007*). Some RAC1-positive cells (A) and clusters of RAC2-positive tumor cells (C)

*Figure 5 continued on next page*

*Figure 5 continued*

correlate with CD34+ tumor cells (white arrows) in the anorectal SCC but not in the skin SCC (**B, D**). All images have been acquired using the same laser parameters and exposure time. DAPI counterstains nuclei in blue. Abbreviation: bv, blood vessel. Scale bars = 20 µm (**A–A''–B–B''**), 10 µm (**C–C''–D–D''**). n = 3 different skin and anorectal tumors tested for each antibody. See also *Figure 5—figure supplement 1*.

The following figure supplement is available for figure 5:

**Figure supplement 1.** Venn diagrams of cell-type specific signatures in various skin SCC.

tumor or YFP+mCherry+ cells isolated from tumors generated from *Tgfbr2* cKO CD34+ SCC cells infected with empty vector (*Figure 6G*). No *Tgfbr2* mRNA was detected by qRT-PCR in YFP+mCherry- cells isolated from tumors generated from *Tgfbr2* cKO CD34+ SCC cells infected with full-length *Tgfbr2* or YFP+mCherry+ cells isolated from tumors generated from *Tgfbr2* cKO CD34+ SCC cells infected with empty vector (*Figure 6G*), indicating that mCherry expression faithfully represented cells in which TGFβ signaling had been restored. Restoration of *Tgfbr2* in *Tgfbr2* cKO CD34+ SCCs dramatically reduced the frequency of mCherry+CD34+ CSCs from 6.5% to 2.5% (*Figure 6F*), demonstrating that loss of TGFβ signaling is a requirement for CSC maintenance in transition zone carcinoma. Importantly, rescue of *Tgfbr2* abolished *Elmo1* mRNA expression in YFP+mCherry+ cells isolated from tumors generated from *Tgfbr2* cKO CD34+ SCC cells infected with full-length *Tgfbr2* compared to YFP$^+$mCherry$^-$CD34$^+$ cells isolated from the same tumor (*Figure 6H*), confirming that the RAC-activating GEF *Elmo1* is a novel target of TGFβ repression.

## Expression of ELMO1 is found in human TGFβ-deficient invasive anorectal SCC

Given that *Tgfbr2*-deficient anorectal SCC expressed the GEF ELMO1, we wondered whether ELMO1 might also be expressed in human anorectal cancers. We performed immunohistochemistry on a series of human anorectal biopsies that ranged from normal mucosa to invasive grade three carcinomas. We found that ELMO1 was expressed in 5 of the 15 of anorectal tumors tested (*Table 1*). Concomitant with this expression was a corresponding loss of phosphorylated SMAD2 within the tumor tissue in 5 out of 6 invasive SCC (*Table 1* and *Figure 7C–E*, *Figure 7—figure supplement 1*). Interestingly, none of the early stage tumors tested (anal intraepithelial neoplasia or SCC in situ) expressed ELMO1, and these specimens stained strongly for nuclear phosphorylated SMAD2 similarly to normal anorectal mucosa (*Figure 7A–B*, *Figure 7—figure supplement 1*). Taken together, these data support a role for ELMO1 in invasive TGFβ-deficient transition zone SCC.

## Knockdown of *Elmo1* diminishes cell migration in vitro, RAC localization at the tumor-stroma border, and reduces markers of invasion in *Tgfbr2*-deficient SCC CD34+ CSCs

To determine whether a reduction in ELMO1 could reduce invasion and metastasis in *Tgfbr2* cKO SCC, we knocked down *Elmo1* in *Tgfbr2* cKO CD34$^+$ SCC cells using two shRNA constructs (*Figure 8*). Treatment of *Tgfbr2* cKO CD34$^+$ SCC cells with *Elmo1* shRNA resulted in 40% (construct #1) or 50% (construct #2) reduction in endogenous *Elmo1* mRNA levels, compared to cells infected with control shRNA (SH02) (*Figure 8A*). Western blot analysis confirmed this reduction at the protein level (*Figure 8B*). To confirm the specificity of the shRNA, we used a hairpin-resistant ELMO1 cDNA (ELMO1*) in which we had introduced three base mutations in the target sequence of the *Elmo1* shRNA #2 without affecting the function of ELMO1 (*Figure 8—figure supplement 1*). We cloned this construct into the pLVX-IRES-mCherry lentiviral vector and infected *Tgfbr2* cKO CD34+ SCC *Elmo1* shRNA expressing cells with ELMO1* or the empty vector. Western blot analysis confirmed that overexpression of the hairpin-resistant ELMO1* construct restores similar level of expression of ELMO1 (*Figure 8C*). We performed an in vitro wound healing assay to show that knocking down *Elmo1* in *Tgfbr2* cKO CD34+ SCC cells affected their ability to migrate and close the wound. This effect was rescued when *Elmo1* knockdown cells expressed the hairpin-resistant ELMO1* construct (*Figure 8D–E* and *Video 1*). We confirmed that the effect in cell migration was not due to a difference in proliferation as measured by flow cytometry (*Figure 8F*).

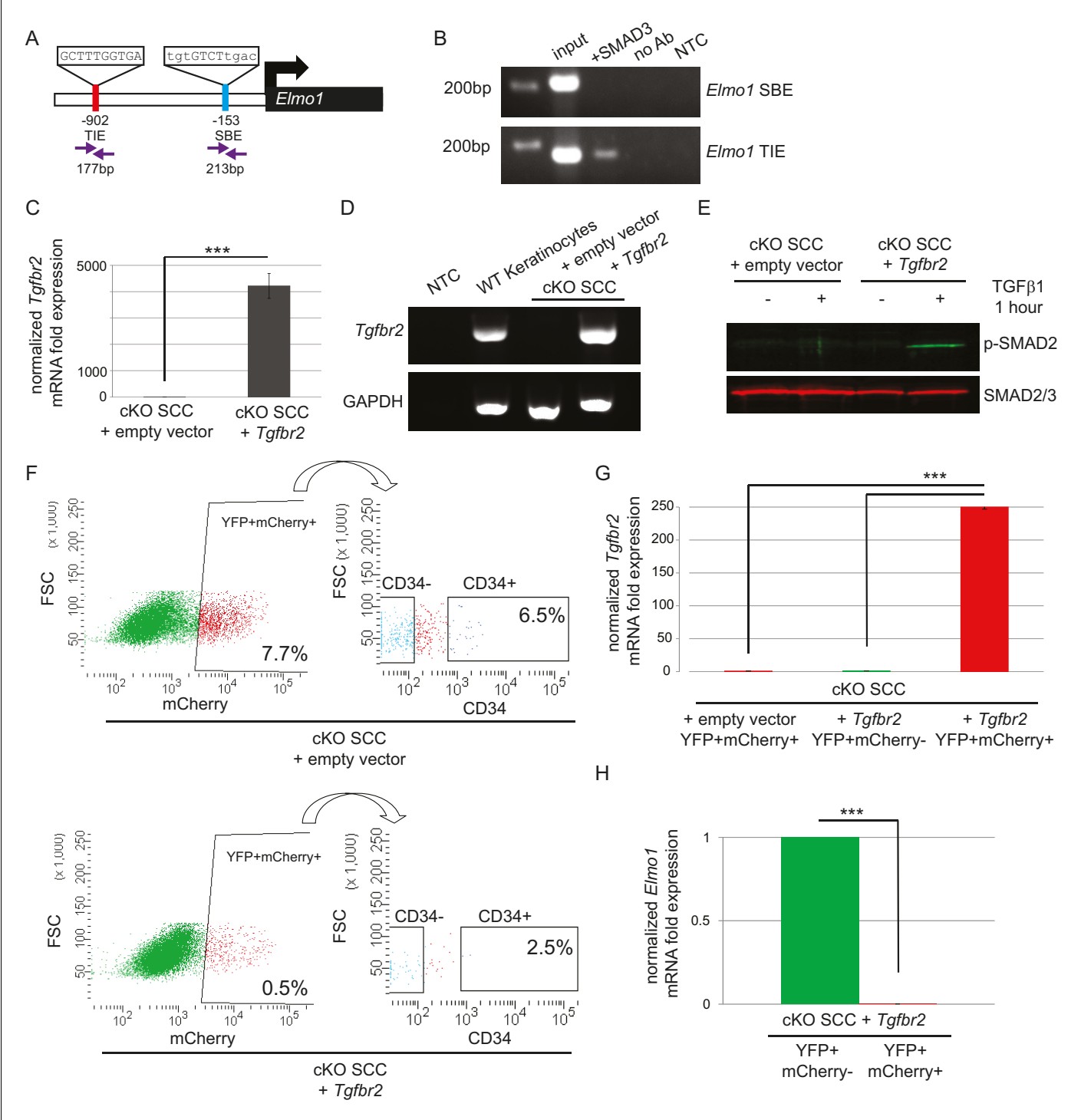

**Figure 6.** The GEF ELMO1 is a novel target of TGFβ signaling via SMAD3. (**A**) Promoter analysis using MatInspector (Genomatrix) revealed two putative SMAD-responsive elements in the *Elmo1* promoter. The consensus SMAD-binding element (SBE) sequence GTCT was identified 153 base pairs (bp) upstream of the *Elmo1* transcriptional start site, and the consensus TGFβ-inhibitory site (TIE) GNNTTGGNGN was identified 902 bp upstream of the *Elmo1* transcriptional start site. Primers were designed to flank these sites (purple arrows). (**B**) Chromatin immunoprecipitation with an anti-SMAD3 antibody was used to isolate DNA fragments that were amplified by the primers designed to flank the *Elmo1* TIE, but not the *Elmo1* SBE, after overexpressing SMAD3 in NIH3T3 cells and treating with TGFβ1 (2 ng/ml) for 24 hr. Non-template (NTC) and no-antibody (no Ab) controls were used to verify the specificity of binding. (**C–E**) Lentiviral infection of *Tgfbr2* cKO CD34+ SCC cells with the full-length *Tgfbr2* gene inserted into the pLVX-IRES-mCherry vector resulted in rescue of *Tgfbr2* mRNA by more than 4000-fold (**C–D**) and phosphorylated SMAD2 (p-SMAD2) in response to

*Figure 6 continued on next page*

*Figure 6 continued*

treatment with TGFβ1 (2 ng/ml) for 1 hr (E), compared to *Tgfbr2* cKO CD34+ SCC cells infected with the empty pLVX-IRES-mCherry vector. No *Tgfbr2* mRNA was detected in *Tgfbr2* cKO CD34+ SCC cells infected with the empty pLVX-IRES-mCherry vector. Data represent the mean ± standard deviation; student's *t*-test, ***p=0.000112. (F–I) Tumors generated from orthotopic transplantation of 100,000 *Tgfbr2* cKO CD34+ SCC cells infected with empty vector or with full-length *Tgfbr2* were dissociated and YFP+mCherry+, YFP+mCherry-CD34+ and YFP+mCherry-CD34− cells were isolated by FACS and subjected to RNA extraction. (F) Approximately 7.7% of the cKO SCC + empty vector total tumor bulk expressed mCherry, whereas 0.5% of the cKO SCC + *Tgfbr2* expressed mCherry at the time of analysis. See also *Figure 6—figure supplement 2*. The frequency of YFP+mCherry+CD34+ cells was significantly reduced in the rescued cKO SCC + *Tgfbr2* tumor. (G) YFP+mCherry+ cells isolated from tumors generated from *Tgfbr2* cKO CD34+ SCC cells infected with full-length *Tgfbr2* expressed *Tgfbr2* mRNA 250-fold over YFP+mCherry- cells isolated from the same tumor or YFP+mCherry+ cells isolated from tumors generated from *Tgfbr2* cKO CD34+ SCC cells infected with empty vector (***p=0.000005). No *Tgfbr2* mRNA was detected by qRT-PCR in YFP+mCherry- cells isolated from tumors generated from *Tgfbr2* cKO CD34+ SCC cells infected with full-length *Tgfbr2* or YFP+mCherry+ cells isolated from tumors generated from *Tgfbr2* cKO CD34+ SCC cells infected with empty vector. (H) Rescue of TGFβRII abolished *Elmo1* mRNA expression in YFP+mCherry+ cells isolated from tumors generated from *Tgfbr2* cKO CD34+ SCC cells infected with full-length *Tgfbr2* compared to YFP+mCherry-CD34+ cells isolated from the same tumor. Data represent the mean ± standard deviation. Asterisks denote statistical significance using two-tailed, unpaired student's *t*-test; p***=9×10$^{-24}$. Three different tumors for each condition have been analyzed.

The following figure supplements are available for figure 6:

**Figure supplement 1.** *Dock2* is not a direct target of TGFβ signaling via SMAD3.

**Figure supplement 2.** Restoration of *Tgfbr2* caused a two-fold delay in tumor formation.

We transplanted these cells orthotopically into the anorectal transition zone of recipient mice and observed tumor formation with no difference in latency. Consistent with this result, we observed that *Elmo1* knockdown does not affect the proportion of CSC CD34+YFP+mCherry+ cells analyzed by FACS (Control SH02: 4.1% compared to *Elmo1* shRNA: 3.3%, *Figure 9—figure supplement 1A*) We isolated epithelial YFP+CD34+ cells from these tumors, using the same FACS strategy as described in *Figure 1*, and observed a 60% reduction in *Elmo1* mRNA compared to CD34+ cells isolated from SH02 tumors, validating in vivo the loss of *Elmo1* (*Figure 9—figure supplement 1B*). Infection of the cells with the pLVX-mCherry vector did not affect the efficiency of the shRNA, as we observed a similar reduction in *Elmo1* mRNA compared to CD34+ cells isolated from SH02 tumors that did not express the mCherry (50% reduction for construct #1 and 60% reduction for construct #2 (*Figure 9—figure supplement 1C*). Using immunofluorescence we confirmed reduction in ELMO1 expression at the protein level in these tumors (*Figure 9—figure supplement 2A*) compared to SH02 tumors. Infection of CD34+ *Elmo1* shRNA SCC cells with the hairpin-resistant ELMO1* construct efficiently restored *Elmo1* mRNA and ELMO1 protein expression in the resulting tumors (*Figure 9—figure supplement 2B''*), indicating that mCherry expression faithfully represented cells in which ELMO1 expression had been restored. Consistent with a migration defect in vitro (*Figure 8D*), we observed a dramatic reduction in RAC1 staining at the tumor-stroma border in tumors with *Elmo1* knockdown (*Figure 9A–A'*), which was rescued in cells expressing the hairpin-resistant ELMO1* construct (*Figure 9A''*).

Because RAC signaling plays a role in tumor invasion, we hypothesized that a number of markers of EMT and/or invasion would be altered upon *Elmo1* knockdown, and indeed observed reduction in the mRNA expression of *Snail*, *αSma*, *Vimentin*, *Zeb2* and *Mmp9* in CD34+ cells isolated from *Tgfbr2* cKO tumors infected with both *Elmo1* shRNA constructs #1 and #2 compared to those infected with SH02 control (*Figure 9B*). The level of mRNA expression of these genes was restored or increased in the cells expressing the hairpin-resistant ELMO1* construct (*Figure 9—figure supplement 2B*). This increase can be explained by the fact that the level of *Elmo1* in the rescued cells was higher than endogenous levels in the SH02 control cells. These results indicate that *Elmo1* reduction alters localization of RAC in *Tgfbr2*-deficient SCC in vivo and reduces the expression of invasive markers in CD34+ CSCs.

## Knockdown of ELMO1 inhibits metastasis in *Tgfbr2*-deficient SCC

To determine whether *Elmo1* knockdown altered tumor progression and metastasis in *Tgfbr2* cKO SCC, we screened the lungs of tumor-bearing mice for YFP+ metastases by FACS. We observed a

**Table 1.** ELMO1 is expressed in human TGFβ-deficient invasive anorectal SCC. Human anorectal tumor biopsies from male and female patients, aged 32–70, were analyzed for phosphorylated (activated) SMAD2 (pSMAD2) and ELMO1 by immunohistochemistry (IHC) (see **Figure 7** and **Figure 7—figure supplement 1**). Loss of pSMAD2 correlated with increased ELMO1 expression in 5 out of 6 SCC samples. Scoring: (++), strong positive staining; (+), positive staining; (-), negative staining. Abbreviations: SCC: squamous cell carcinoma; AIN3: Anal intraepithelial neoplasia (early stage tumor).

| Diagnosis | Sex | Age | pSMAD2 | ELMO1 | Pictures |
|---|---|---|---|---|---|
| normal anorectal mucosa | M | 58 | ++ | − | *Figure 7A* |
| SCC in situ | M | 47 | ++ | − | *Figure 7—figure supplement 1A* |
| SCC in situ + focus of invasion | M | 37 | ++ | − | |
| SCC in situ with microinvasion | M | 69 | ++ | − | |
| SCC in situ + invasive SCC | F | 70 | ++ and − | − | |
| AIN3 | F | 32 | ++ | − | *Figure 7B* |
| AIN3 | M | 52 | ++ | − | |
| invasive SCC + AIN3 | M | 61 | ++ and − | + | |
| invasive SCC + AIN3 | M | 52 | ++ and − | + | *Figure 7—figure supplement 1B* |
| invasive SCC grade 2 | M | 51 | ++ | − | |
| invasive SCC grade 2 | M | 59 | ++ | − | |
| invasive SCC grade 2 | F | 51 | ++ | − | |
| invasive SCC grade 3 | M | 53 | ++ | − | |
| invasive SCC grade 3 | F | 55 | ++ and − | + | *Figure 7C* |
| invasive SCC grade 3 | F | 53 | + and − | ++ | *Figure 7D* |
| invasive SCC grade 3 | M | 48 | − | ++ | *Figure 7E* |

dramatic overall reduction in the number of YFP+ cells in the lungs of mice orthotopically transplanted with *Tgfbr2* cKO CD34+ cells infected with *Elmo1* shRNA knockdown compared to those infected with SH02 control (n = 6 mice in each condition) (**Figure 10A–B**). We detected 1.4% YFP+ cells in the lungs of mice transplanted with the SH02 control cells, whereas we detected a dramatic reduction to 0.4% YFP+ cells in mice transplanted with the ELMO1 shRNA#1 and failed to detect any YFP+ cells in the lungs of mice transplanted with ELMO1 shRNA #2. We serially sectioned and stained the whole lungs from additional tumor-bearing mice, and observed YFP+ metastatic nodules in mice transplanted with SH02 cells, but never in mice transplanted with *Elmo1* shRNA-infected cells (**Figure 10C**), validating the sensitivity of FACS in providing a quantitative method to screen for lung metastases. Taken together, these data demonstrate that upregulation of the GEF ELMO1 is required for *Tgfbr2*-deficient SCC CD34+ CSCs to metastasize.

## Discussion

Transition zones are found throughout the body and are preferential sites for malignant tumor formation. Despite the clinical importance of these tumors, the cellular and molecular mechanisms defining their growth and progression remain unknown. Here, we have taken advantage of a mouse model that develops spontaneous tumors at their anorectal transition zone upon loss of TGFβ signaling, which recapitulates invasive human anogenital squamous cell carcinoma, to define the cells and

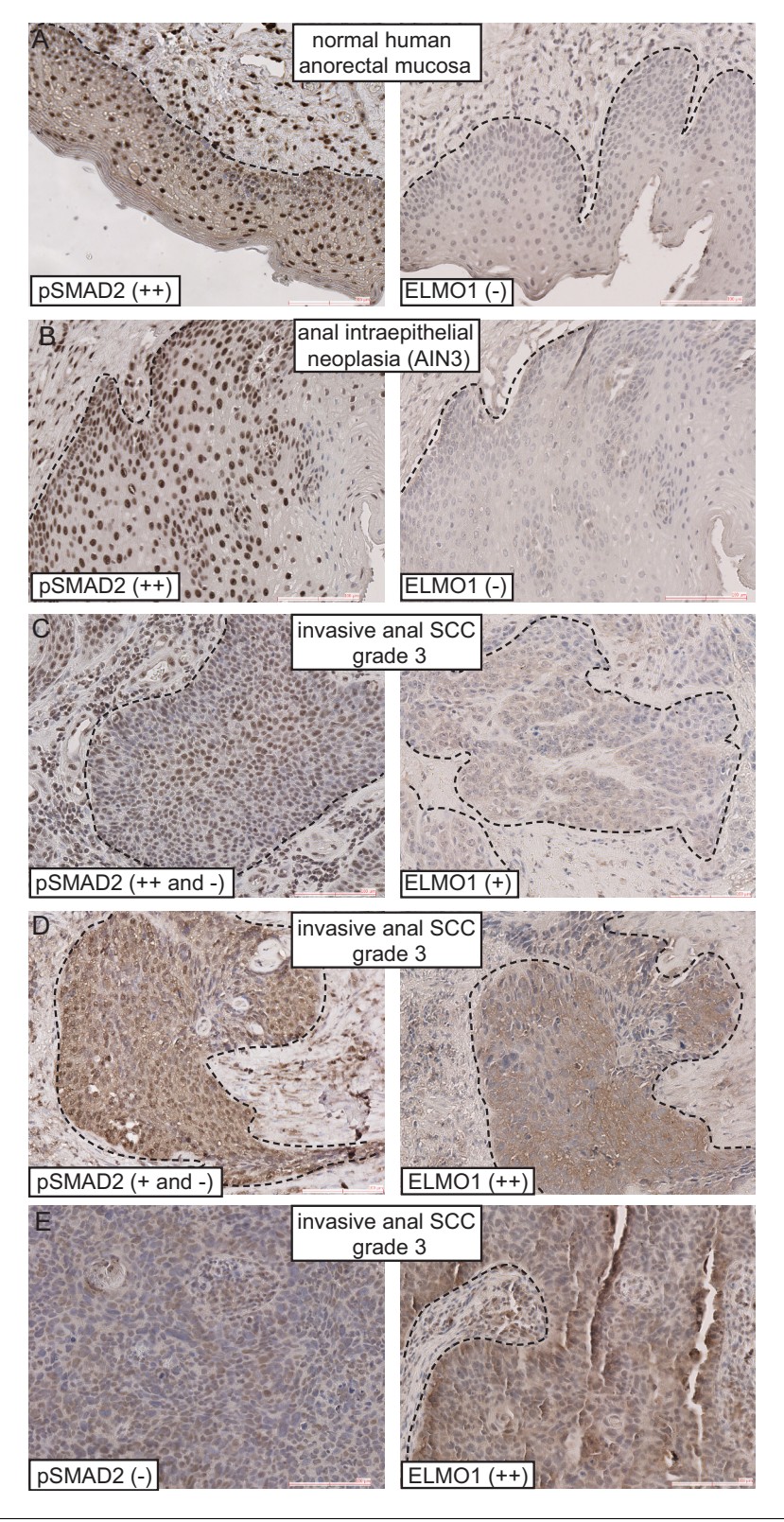

**Figure 7.** ELMO1 is expressed in human TGFβ-deficient invasive anorectal SCC. Human anorectal tumor biopsies were analyzed for phosphorylated (activated) SMAD2 (pSMAD2) and ELMO1 by immunohistochemistry (IHC). See also *Figure 7—figure supplement 1* for additional tumor biopsy sections and antibody controls. Examples of IHC staining from three invasive anal SCC grade 3 (**C–E**) show reduced or absent nuclear pSMAD2 staining in contrast
*Figure 7 continued on next page*

*Figure 7 continued*

to normal anorectal mucosa (**A**) and early stage tumor, anal intraepithelial neoplasia (**B**). Loss of pSMAD2 correlated with increased ELMO1 expression in 5 out of 6 SCC samples (see *Table 1*). Scoring: (++), strong positive staining; (+), positive staining; (−), negative staining. Dashed lines delineate the epidermis from dermis in (**A**) and delineate the tumor from stroma in (**B–E**). Hematoxylin counterstains nuclei in blue. Scale bars = 100 μm.

The following figure supplement is available for figure 7:

**Figure supplement 1.** ELMO1 is expressed in human TGFβ-deficient invasive anorectal SCC.

mechanisms by which these tumors grow and to identify the signaling pathways upon which they rely when undergoing metastasis.

We have shown that epithelial CD34+ cells of *Tgfbr2* cKO SCC, which are clonogenic in vitro, able to self-renew in vivo, and generate differentiated progeny in serial transplantation assay are potential CSCs. We have further shown that restoration of TGFβRII in *Tgfbr2* cKO SCC reduced the frequency of these CD34+ CSCs, demonstrating that loss of TGFβ signaling is a requirement for CSC maintenance in transition zone carcinoma. This is in accordance with what has been described in *Hras* mutated DMBA-induced skin SCC, where the frequency of CD34+ CSCs was 10 fold higher in *Tgfbr2* cKO mice than in WT mice (*Schober and Fuchs, 2011*). However, we find little overlap between gene signatures of CD34+ CSCs from *Tgfbr2* cKO transition zone SCC and from *Tgfbr2* cKO + *Hras* backskin SCC. Upregulated genes in the CD34+ CSC signature of *Tgfbr2* cKO + *Hras* backskin SCC include those involved in cell cycle progression and DNA repair (*Schober and Fuchs, 2011*), whereas the CD34+ CSC signature in our *Tgfbr2* cKO anorectal SCC is primarily comprised of genes involved in invasion, metastasis, and activation of the RAC signaling pathway. Interestingly, tumor cells with low CD34 expression displayed greater tumorigenicity than cells with high CD34 expression in the *Tgfbr2* cKO + *Hras* backskin SCC (*Schober and Fuchs, 2011*), whereas we observed a dramatic enrichment for tumorigenicity in CD34+ SCC cells from transition zone tumors. We found very few genes in common between *Tgfbr2*-deficient anorectal CSCs and CSCs from DMBA-induced *Hras* skin SCC from WT mice or mice deficient in FAK, which is expected as these mice are refractory to DMBA-induced skin SCC compared to the accelerated skin SCC development in *Tgfbr2* deficient mice.

We found more overlap between transition zone SCC CSCs and backskin SCC CSCs when we compared the gene profile of *Tgfbr2*-deficient anorectal CSCs to CSCs from *Hras* + VEGF overexpressing backskin SCC. When mice were administered the DMBA skin carcinogenesis protocol while overexpressing VEGF, the resulting skin tumors were more aggressive and the stemness profile of CD34+ CSCs was qualitatively altered compared to wild-type mice (*Beck et al., 2011*). We found 45 genes commonly upregulated between *Tgfbr2*-deficient anorectal CD34+ CSCs and *Hras* + VEGF overexpressing SCC CD34+ skin CSCs, and 40 genes that were upregulated in *Tgfbr2*-deficient anorectal CD34+ CSCs but downregulated by VEGF in skin CSCs. Three of these genes (*Rgs16*, *Rassf3* and *Fam13a*) are members of the RAC/RHO/RAS pathway, and dysregulated *Rgs16* has been associated with poor prognosis in colorectal cancer (*Miyoshi et al., 2009*). Our gene expression analysis is evidence that the requirements of CSCs within distinct tumor types vary, and that their growth and invasive properties are heterogeneous and context-dependent. Furthermore, our data show that deregulated expression of RAC1 and RAC2 is evident in *Tgfbr2* deficient transition zone tumors but not in skin SCC in various genetic backgrounds analyzed. While we found three RAC/RHO/RAS family members which were common between *Tgfbr2*-deficient anorectal CSC and genes up- or downregulated in DMBA-induced backskin CSCs in the aggressive VEGF gain-of-function mouse model, none of these three genes are RAC-activating GEFs. This suggests that the mechanism linking loss of TGFβ signaling with de-repression of RAC-activating GEFs is unique to CSCs within transition zone tumors, which may explain their particular aggressiveness and malignancy.

Our findings identify a novel mechanism whereby loss of TGFβ signaling in transitional epithelial carcinoma drives an invasive and metastatic program specifically in cancer stem cells. We established the metastatic gene signature of CD34+ CSCs in *Tgfbr2* cKO anorectal SCC and found that the GEF ELMO1 is a direct target of SMAD3. While more than 70 GEFs have been described to date (*Parri et al., 2010*), ELMO1 has been specifically identified as oncogenic in a variety of studies

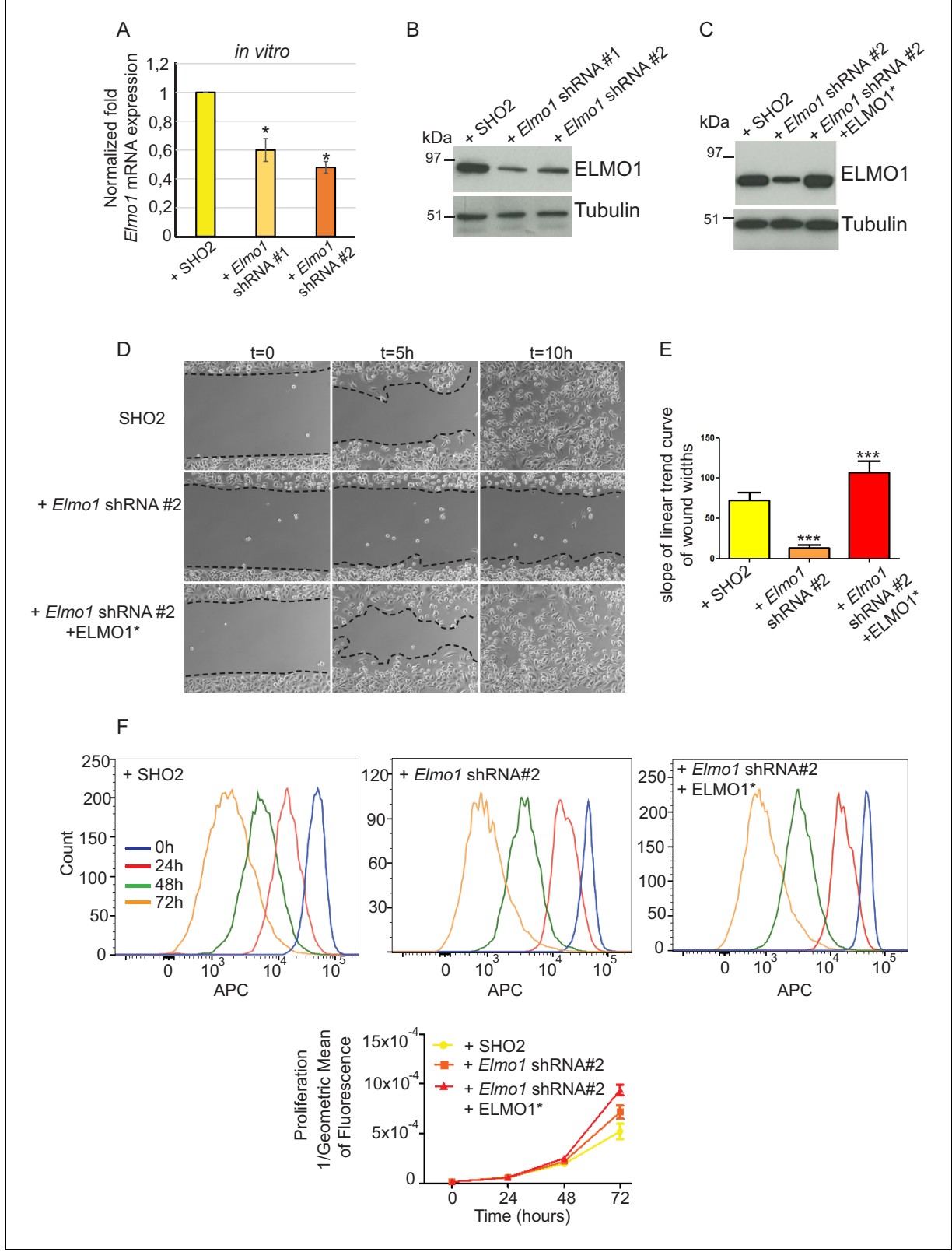

**Figure 8.** Knockdown of *Elmo1* in vitro affects cell migration. (**A**) shRNA knockdown of *Elmo1* in cKO SCC CD34+ cells with two constructs resulted in 40–50% reduction in endogenous *Elmo1* mRNA expression compared to cKO SCC CD34+ cells infected with control shRNA (SH02). Asterisks denote statistical significance using paired-sample Wilcoxon Signed Rank test; p*=0.0350 (*Elmo1* shRNA#1), p*=0.0355 (*Elmo1* shRNA#2). (**B**) Western blot analysis confirmed the reduction in endogenous ELMO1 protein in cKO SCC cells compared to cells infected with control shRNA. (**C**) Western blot

*Figure 8 continued on next page*

*Figure 8 continued*

analysis confirmed that overexpression of the hairpin-resistant ELMO1 construct (ELMO1*) in the *Elmo1* shRNA cKO SCC cells restored expression of *Elmo1*. See also **Figure 8—figure supplement 1**. (**D–E**) Wound healing assay in vitro showed that knockdown of *Elmo1* in cKO SCC cells impaired their ability to migrate 5 hr and 10 hr after wounding. This effect was rescued when the hairpin-resistant Elmo1 construct (ELMO1*) is expressed. See also **Video 1**. (**E**) Quantification of the wound healing assay showing the slope of linear trend curve of wound widths normalized to SH02 control. For each construct three different cell lines have been tested and the experiments have been repeated five times. Asterisks denote statistical significance using two-tailed, unpaired student's *t*-test; p***<0.0001 (*Elmo1* shRNA#2), p***=0.0002 (*Elmo1* shRNA#2 + ELMO1*). (**F**) *Elmo1* knockdown did not affect cell proliferation. Histograms show the fluorescence of the dye eFluor 670 in the APC channel at 0 hr, 24 hr, 48 hr and 72 hr. Decrease of the fluorescence reflected proliferation by the dilution of the dye over time. Three separate experiments have been done and the mean of the geometric mean for each sample is represented in the graph. There was no statistical difference in cell proliferation between samples calculated by two-way ANOVA and Bonferroni post tests.

The following source data and figure supplement are available for figure 8:

**Source data 1.** Values and statistics for *Figure 8A* using paired-sample Wilcoxon Signed Rank test.
**Source data 2.** Values and statistics for *Figure 8E* using two-tailed, unpaired student's *t*-test.
**Source data 3.** Values and statistics for *Figure 8F* using two-way ANOVA and Bonferroni post tests.
**Figure supplement 1.** Expression of the hairpin-resistant ELMO1* construct.

(**Lazer and Katzav, 2011**; **Jarzynka et al., 2007**; **Li et al., 1706**; **Jiang et al., 2011**; **Ungefroren et al., 2011**). Upregulation of GEFs is a frequent mechanism of increased RAC activity and cancer cell invasion and metastasis (**Lazer and Katzav, 2011**; **Ensign et al., 2013**). We have shown here that loss of TGFβ-mediated repression is one mechanism that may lead to this upregulation. There have been many reports of RAC GTPases functioning downstream of TGFβ signaling (**Atfi et al., 1997**; **Meriane et al., 2002**; **Faherty et al., 2013**; **Brown et al., 2008**), but never in the context of elevated GEF expression as a result of loss of TGFβ signaling. In pancreatic ductal adenocarcinoma cells that retain TGFβ signaling, RAC1 regulates SMAD2 and SMAD3 at the post-transcriptional level, shifting TGFβ from a tumor-suppressive role to a tumor-promoting role (**Ungefroren et al., 2011**), suggesting that the relationship between the TGFβ and RHO/RAC GTPase pathways is complex and context-dependent but nevertheless central.

TGFβ signaling is involved in many cellular processes and participates in oncogenesis in several types of cancer, especially aggressive cancers such as pancreatic (**Jaffee et al., 2002**), colon (**Jones et al., 2008**), and oral carcinomas (**Sivadas et al., 2013**). This participation may be through a loss of activity and loss of tumor suppressive function. However, TGFβ has a complex role in tumor progression. It can act as a tumor promoter or suppressor depending on the tumor type and stage. Identifying the consequences of TGFβ signaling alterations in cancer and the players in tumor progression and metastasis for each type of cancer is a prerequisite to derive appropriate therapeutic tools. RAC inhibitors exist, but are poorly specific. Identification of key downstream components of TGFβ signaling, such as ELMO1, offers new perspectives and opportunities.

In conclusion, this is the first report of a mechanism connecting loss of TGFβ signaling with loss of repression of a RAC-activating GEF. This mechanism may sustain invasion and metastasis in aggressive cancers that lack TGFβ signaling.

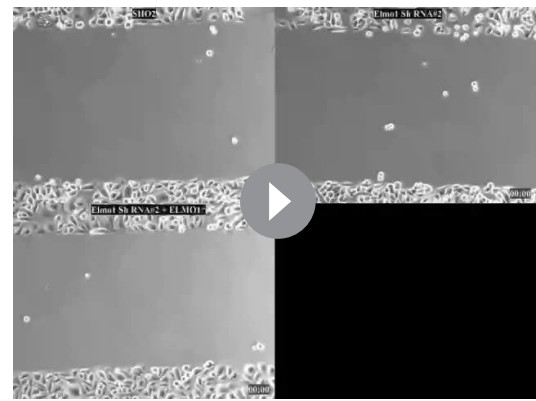

**Video 1.** In vitro wound healing assay showing the migration of cKO SCC SHO2 control, knockdown of *Elmo1* and the hairpin-resistant Elmo1 construct. Images were taken every 10 min for 10 hr.

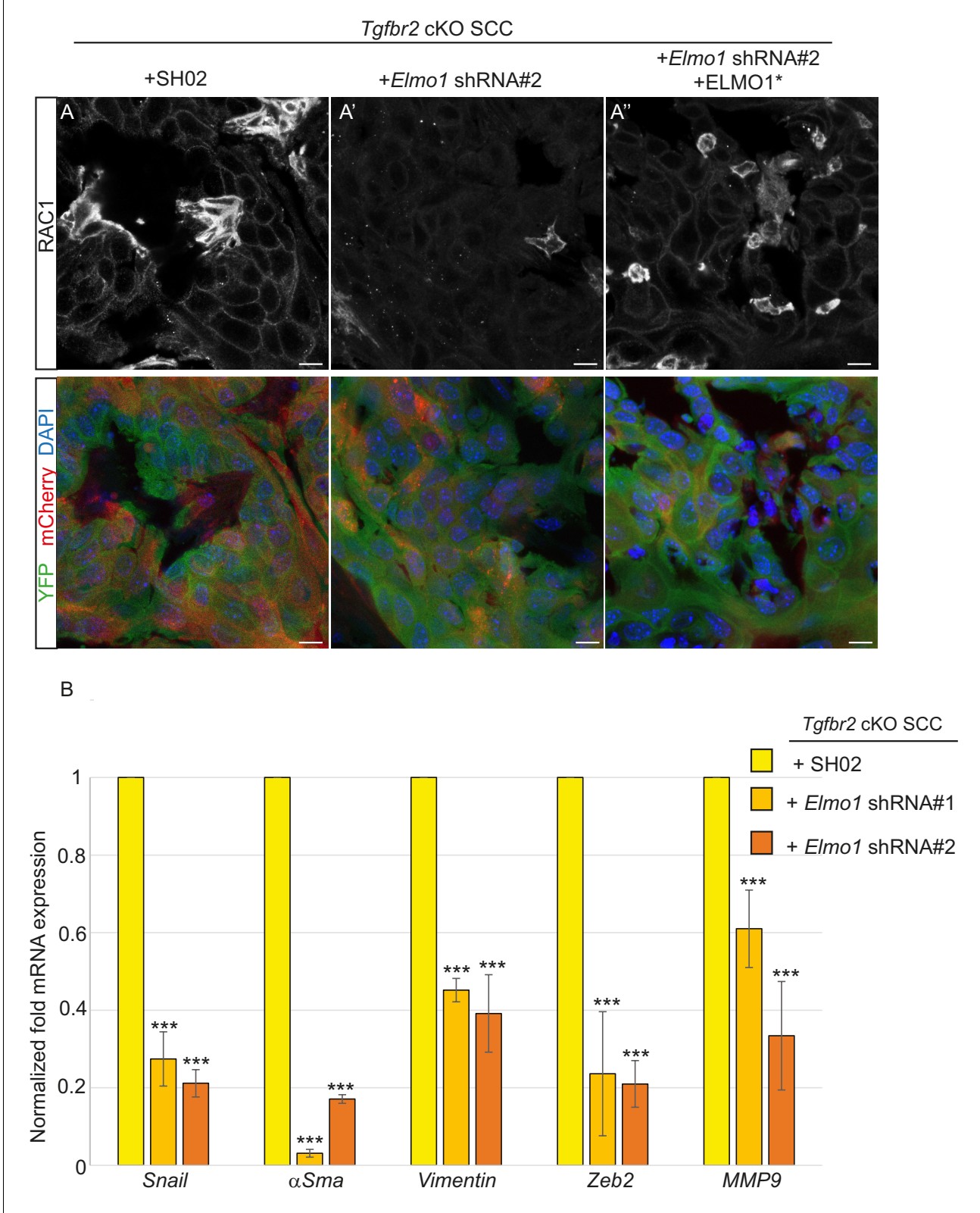

**Figure 9.** Knockdown of *Elmo1* diminishes RAC1 expression and markers of invasion in *Tgfbr2*-deficient SCC. (**A**) Immunofluorescence staining with antibodies against YFP and RAC1 revealed a reduction in RAC1 staining at the tumor-stroma border in *Tgfbr2* cKO tumors with knockdown of *Elmo1* (**A'**), compared to SH02 control tumors (**A**). Tumors expressing the hairpin-resistant ELMO1* construct show restoration of RAC1 compared to tumors with *Elmo1* knockdown (**A''**). All pictures have been taken at the same exposure time in the RAC1 channel. Three tumors for each condition have been

*Figure 9 continued on next page*

*Figure 9 continued*

analyzed. DAPI counterstains all nuclei in blue. Scale bars = 10 µm. (**B**) qPCR analysis of genes implicated in EMT, invasion and metastasis revealed a significant reduction in their mRNA expression in CD34+ cells isolated from *Tgfbr2* cKO tumors with knockdown of *Elmo1* compared to SH02 control tumors (see *Figure 9—figure supplement 1* and *2*). Data represent the mean ± standard deviation. Asterisks denote statistical significance using two way-ANOVA and Bonferroni post tests to compare each *Elmo1* shRNA to SHO2 control. All p-values are <0.001.

The following source data and figure supplements are available for figure 9:

**Source data 1.** Values and statistics for *Figure 9B* using two way-ANOVA and Bonferroni post tests.
**Figure supplement 1.** Knockdown of *Elmo1* in vivo.
**Figure supplement 2.** Knockdown of *Elmo1* diminishes markers of invasion in *Tgfbr2-* deficient SCC.

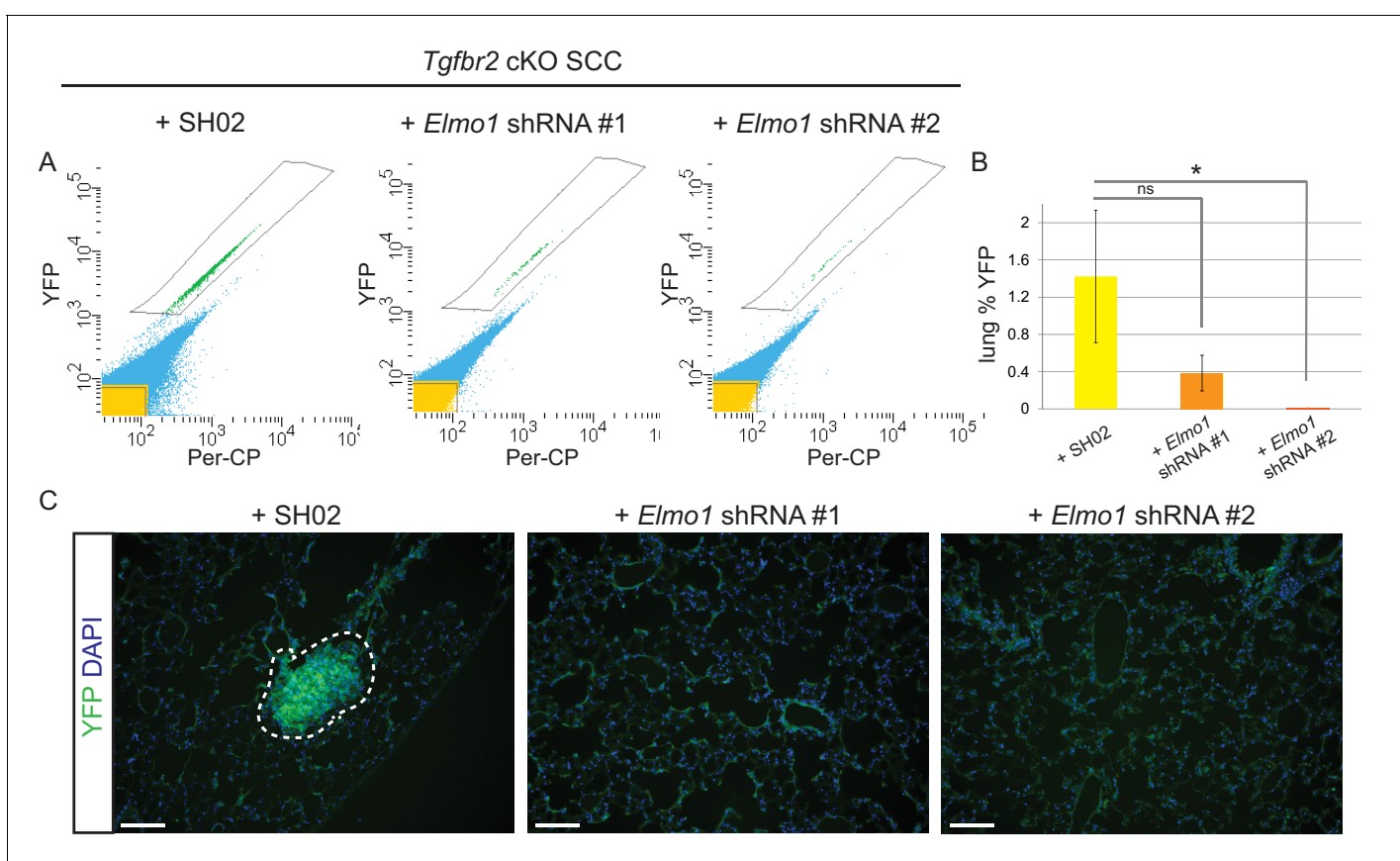

**Figure 10.** Knockdown of *Elmo1* inhibits *Tgfbr2*-deficient SCC metastasis. (**A–B**) Whole lungs from mice bearing tumors from orthotopic transplantation of *Tgfbr2* cKO CD34+ SCC cells infected with SH02 control and *Elmo1* shRNA were dissociated and total YFP+ cells were quantified by FACS (n = 6 different mice analyzed for each construct). Per-CP was used to exclude auto-fluorescent cells. Data represent the mean ± standard deviation. Asterisks denote statistical significance using two-tailed, unpaired student's *t*-test; *p=0.0318. (**C**) Upon serial section of entire lungs, YFP+ lung metastases were only observed microscopically in mice bearing tumors from orthotopic transplantation of *Tgfbr2* cKO CD34+ SCC cells infected with SH02 control, but not infected with *Elmo1* shRNA#1 or #2. DAPI counterstains nuclei in blue. Scale bars = 100 µm.

The following source data is available for figure 10:

**Source data 1.** Values and statistics for *Figure 10B* using two-tailed, unpaired student's *t*-test.

## Materials and methods

### Mice

The conditional knockout *Tgfbr2^flox/flox* x *K14-Cre* mouse model (*Guasch et al., 2007*; *Leveen et al., 2002*) has been derived in a pure C57BL/6N background and backcrossed into a mouse reporter containing an Enhanced Yellow Fluorescent Protein gene (eYFP) inserted into the *Gt(ROSA)26Sor* locus (*Srinivas et al., 2001*) and called *R26R-eYFP^flox-STOP-flox* (Jackson Laboratory) (*McCauley and Guasch, 2013*; *McCauley et al., 2014*). Control mice were either *Tgfbr2^flox/flox* x *R26R-eYFP^flox-STOP-flox* or *Tgfbr2 ^+/+* x *R26R-eYFP^flox-STOP-flox* x *K14-Cre,* all in a C57BL/6N background. Transplantation assays were carried out in homozygous *Nu/Nu* female mice, approximately six to eight weeks old, as previously described (*McCauley and Guasch, 2013*). The chemical mutagen 7,12-dimethyl-benz[a] anthracene (DMBA) was administered topically to the backskin of cKO mice for 16 weeks as previously described (*Guasch et al., 2007*). Mice are housed in a sterile barrier facility as previously described (*McCauley and Guasch, 2013*). All experiments were approved by the Cincinnati Children's Hospital Research Foundation Institutional Animal Care and Use Committee (protocol # 1D10087) and in agreement with European and national regulation (protocol # 4572) and carried out using standard procedures. The identification of each allele was performed by PCR on DNA extracted from clipping the ear of the mice as previously described (*Guasch et al., 2007*; *McCauley et al., 2014*).

### Fluorescence-activated cell sorting (FACS)

Tumors were dissociated into a single cell suspension according to protocol established in our lab (*McCauley and Guasch, 2013*) and CD34+ cancer stem cells were isolated as described at Bio-protocol (*McCauley and Guasch, 2017*). Tumors were dissociated into a single cell suspension according to protocol established in our lab (*McCauley and Guasch, 2013*). Cells in suspension were labeled with the following antibodies at the dilutions indicated: PE-Cy7 conjugated to rat-anti-mouse CD11b (BD Pharmingen, 1/200, RRID:AB_2033994), PE-Cy7 conjugated to rat-anti-mouse CD31 (BD Pharmingen 1/100, RRID:AB_10612003), PE-Cy7 conjugated to anti-mouse CD45 (eBioscience 1/200, RRID:AB_469625), PE conjugated to CD49f, (BD Biosciences, 1/50, RRID AB_396079), Pacific Blue conjugated to CD29 (Biolegend, 1/100, RRID:AB_2128079), biotin conjugated to anti-mouse CD34 (eBioscience, 1/50, RRID:AB_466426) and APC conjugated to streptavidin (BD Pharmingen, 1/200, RRID:AB_10050396). Immediately before sorting, cells were incubated with 7-amino-actinomycin D (7-AAD, eBioscience, 20 µl 0.05 mg/ml stock per $10^6$ cells) to exclude dead cells. Tumor cell populations were sorted for RNA extraction, tissue culture or transplantation using sterile practices using a FACS Aria II (BD Biosciences) and FACSDiva software (BD Biosciences) in the Research Flow Cytometry Core at CCHMC. Cells isolated for RNA extraction were collected directly into cell lysis buffer containing beta-mercaptoethanol, vortexed and stored at −80°C until RNA extraction. Cells isolated for tissue culture were collected in epithelial cell culture media (*Nowak and Fuchs, 2009*) (E media) containing 0.05 mM calcium, centrifuged at 1000 RPM for 5 min at 4°C, resuspended in E media containing 0.3 mM calcium, and plated on a feeder layer of irradiated fibroblasts. Cells isolated for transplantation were collected in E media without serum, centrifuged at 1000 RPM for 5 min at 4°C, resuspended in 30% Matrigel and transplanted as previously described (*McCauley and Guasch, 2013*).

### Histological analysis

Tumors were dissected and portions of each tumor were processed for embedding in paraffin as well as embedding in OCT. Pieces of tumor were fixed in formalin for 24 hr at 4°C, then dehydrated and embedded in paraffin in the Pathology Core Facility at CCHMC. Deparaffined sections were then rehydrated and stained with antibodies or Hematoxylin and Eosin in the Pathology Core at CCHMC. Alternatively, using a protocol optimized to preserve YFP expression, pieces of tumor were fixed in 4% paraformaldehyde for 24 hr at 4°C, then washed thoroughly in 1x PBS and soaked in 30% sucrose at 4°C for 24 hr, then incubated in a slurry of 2:1 fresh 30% sucrose:OCT at 4°C for 24 hr, then embedded in OCT compound (Tissue-Tek, Sakura, Torrance, CA) and stored at −80°C as previously described (*McCauley and Guasch, 2013*).

## Immunostainings and antibodies

Deparaffined tissue sections (5 µm) were subjected to antigen retrieval and immunostaining as previously described (*Tompkins et al., 2009*). Frozen tissue sections (10 µm) were subjected to immunofluorescence labeling as previously described (*Runck et al., 2010*). Primary antibodies against the following proteins were used at the dilution indicated: green fluorescent protein, conjugated to Alexa Fluor 488 (Invitrogen, 1/1,000); α6 integrin/CD49f (BD Biosciences, 1/100, RRID:AB_396079), β1-integrin/CD29 (Millipore, 1/100, RRID:AB_2128202), biotin conjugated to rat-anti-mouse CD34 (eBioscience, 1/50, RRID:AB_466426), Keratin-5 (Seven Hills Bioreagents, Rabbit 1/250 or Guinea Pig 1/5,000), ELMO1 (Abcam, 1/100, for immunofluorescence, Sigma, 1/50, RRID:AB_1848128 for immunohistochemistry), RAC1 (Cell Signalling, 1/100 and Santa-Cruz, 1/50, RRID:AB_2238100), RAC2 (Millipore, 1/100, RRID:AB_2176134), pSMAD2 (Cell Signaling Technology, Danvers, MA, 1/100, RRID:AB_331673). 4′,6-diamidino-2-phenylindole (DAPI) was utilized as a marker of cell nuclei (Sigma Chemical Co., St. Louis, MO, 1/5,000). Secondary antibodies conjugated to Alexa Fluor 488 or 540 or 649 (Jackson ImmunoResearch, West Grove, PA) were used at a dilution of 1/1,000. For immunohistochemistry, slides were stained with the ABC kit (Vector Laboratories, Burlingame, CA) and counterstained with nuclear fast red (Sigma Chemical Co., St. Louis, MO, USA) according to manufacturers' instructions. Confocal images were acquired by capturing Z-series with 0.3 µm step size on a Zeiss LSM 880 laser scanning confocal microscope. Images in different focal planes were combined using the Zen software.

## Isolation of primary cells and cell culture

Tumor cells were sorted by FACS and CD34− and CD34+ epithelial populations were collected in E media containing 0.05 mM calcium. Cells were centrifuged at 1000 RPM for 5 min at 4°C, resuspended, and plated at equal densities on a feeder layer of irradiated mouse embryonic fibroblasts (MEFs) in E media containing 0.3 mM calcium. MEFs were isolated from wild-type CD-1 mice at embryonic day 13.5 and cultured in DMEM containing 10% serum and 1% penicillin-streptomycin. After expanding MEFs, confluent plates were trypsinized with 0.05% Trypsin-EDTA (Gibco) and irradiated with 60Gy by the CCHMC Comprehensive Cancer Core Facility. Irradiated MEFs were replated at 100% confluency in DMEM containing 10% serum and 1% penicillin-streptomycin one day before sorting and plating tumor cells. On the day of the sort, the media on the irradiated MEFs was changed to E media containing 0.3 mM calcium. Clones begin to appear after 7–10 days of culture, and were passaged by transferring individual clones of cells on Whatman paper to a new plate on a feeder layer of irradiated MEFs. After the third passage, CD34+ SCC cells were grown on plastic without feeders in E media containing 0.05 mM calcium. These cells have been tested mycoplasma free. Wild-type anal keratinocytes were isolated from newborn C57BL/6 mice at postnatal day one by dissecting the anal canal, dissociating epidermis from dermis by incubating in dispase overnight at 4°C, extracting keratinocytes using 0.12% Trypsin-EDTA diluted in versene containing 0.1% glucose, and plating on a feeder layer of irradiated MEFs in E media containing 0.3 mM calcium as described above. Keratinocytes were passaged to a new feeder layer of irradiated MEFs in E media containing 0.3 mM calcium once confluent. After the third passage, wild-type anal keratinocytes were grown on plastic in E media containing 0.05 mM calcium.

## Western blot

Proteins were detected by Western blotting as previously described (*McNairn et al., 2013*). Briefly, cells were lysed and proteins were separated by SDS-PAGE, transferred to nitrocellulose membranes, and subjected to immunoblotting using antibodies to the following proteins at the dilutions indicated: p-Smad2 (Cell Signaling Technology, 1/1,000, RRID:AB_331673), c-Myc (Cell Signaling Technology, 1/1,000, RRID:AB_2151827), Smad2/3 (BD Biosciences, 1/500, RRID:AB_398161), Keratin 8 (NICHD Developmental Studies Hybridoma Bank maintained by the University of Iowa, 1/1000, RRID:AB_531826), β-actin (Sigma, 1/2,000, RRID:AB_476744), RAC1 (Cell Signaling, 1/2000), ELMO1 (AbCam, 1/2000), α-Tubulin (Sigma, 1/5000, RRID:AB_477593), GAPDH (Santa Cruz, 1/5000, RRID: AB_477593). HRP-coupled secondary antibodies were used at 1/2,000 in 5% nonfat milk, and IRDye-conjugated secondary antibodies (Li-COR, Lincoln, NE) were used at 1/10,000 in 5% nonfat milk. Immunoblots were developed using standard ECL (Amersham) and Luminata TM crescendo and

classico (Millipore) as previously described (*McNairn et al., 2013*) or the Odyssey CLx Infrared Imaging System (Li-COR, Lincoln, NE).

## Real-time PCR

Total RNA was isolated using a Qiagen Rneasy Micro Kit and used to produce cDNA using the Maxima first strand cDNA synthesis kit (Fermentas, San Jose, CA). Reverse transcription (RT) reactions were diluted to 10 ng/µl and 1 µl of each RT was used for real-time PCR. Real-time PCR was performed using the CFX96 real-time PCR System, CFX Manager Software and the SsoFast EvaGreen Supermix reagents (Biorad, Hercules, CA) or StepOne Plus real-time PCR system and the Power Sybr Green PCR Master Mix reagents (Applied Biosystems, Grand Island, NY). All reactions were run in triplicate and analyzed using the $\Delta\Delta$CT method with relative expression normalized to *Gapdh*.

Primers used:

*Gapdh*

|  | F | CGTAGACAAAATGGTGAAGGTCGG |
|---|---|---|
|  | R | AAGCAGTTGGTGGTGCAGGATG |

*CD34*

|  | F | ACCACAGACTTCCCCAACTG |
|---|---|---|
|  | R | CGGATTCCAGAGCATTTGAT |

*Ctss*

|  | F | TGCTAGTTATTGCTCTTACCCAG |
|---|---|---|
|  | R | GTAACTACACATTGATCACGACAC |

*Fbn1*

|  | F | ATTGTTCACCGAGTCGATCTG |
|---|---|---|
|  | R | ACGAGAAGCCTGAGAAAGTG |

*Spp1*

|  | F | TGCACCCAGATCCTATAGCC |
|---|---|---|
|  | R | CTCCATCGTCATCATCATCG |

*Mmp9*

|  | F | TCCGTGTCCTGTAAATCTGC |
|---|---|---|
|  | R | CTTTTCCTAGCCCAGTCACTAAG |

*TGFb2*

|  | F | TTTCTGCGTCAGTGTGAGTC |
|---|---|---|
|  | R | CTTTTCCTAGCCCAGTCACTAAG |

*Rac2*

| | | |
|---|---|---|
| | F | CACAGCCCACACGACAG |
| | R | CACACGGAGAAACAGCAATTC |

*Rhoh*

| | | |
|---|---|---|
| | F | CGATCACCTTTTCTACACCCTG |
| | R | CATACAACCCCTCTACAGTGC |

*Rhoj*

| | | |
|---|---|---|
| | F | ATATGCTGGTGAGGTGTTGG |
| | R | AAGACATGAACTAAGGCCACC |

*Vav1*

| | | |
|---|---|---|
| | F | CCATGAACTGTCCTCACCAG |
| | R | CATCTCTGGGCTTTATCCTGG |

*Dock2*

| | | |
|---|---|---|
| | F | TCGGTGGAGAACTTTGTGAG |
| | R | ACGGTTGTCTTTGCTCTCATC |

*Elmo1*

| | | |
|---|---|---|
| | F | ACTTTGGTCTCACTTGTAGCAG |
| | R | CAGTGTGATAGAGGGATTGGTC |

*Elmo2*

| | | |
|---|---|---|
| | F | GATACTTCCCCTTGCCTCAG |
| | R | GCTTCCTGAGACCTACAATGG |

*Tgfbr2*

| | | |
|---|---|---|
| | F | GCAAGTTTTGCGATGTGAGA |
| | R | TCCGTGTTGTGGTTGATGTT |

*Snail*

| | | |
|---|---|---|
| | F | CTCCTACCCCTCAGTATTCATG |
| | R | AGGGAGGTAGGGAAGTGG |

*αSma*

| | F | GTGAAGAGGAAGACAGCACAG |
|---|---|---|
| | R | GGGAGTAATGGTTGGAATGGG |
| *Vimentin* | | |
| | F | ATGGACAGGTGATCAATGAGAC |
| | R | CAGTAAAGGCACTTGAAAGCTG |
| *Zeb2* | | |
| | F | GCAACATACTCTTTCTCCCCAG |
| | R | TCTGAGCCTTCCTGTGAAAAG |

## RNA-Seq

All genomic analysis was performed in GeneSpring NGS. Samples were sequenced using the HiSeq 2000 (Illumina, CA) with 50 bp, single-end reads. Following primer and barcode removal, sequences were aligned to the mm9 mouse genome using Ensembl transcripts. Following alignment, reads were quantified to generate computing reads per kilobase per million reads (RPKM), then normalized using the DESeq algorithm and baselining to the median of all samples. We applied a filter, requiring at least 20 reads in at least one of the four samples. We generated a list of differentially regulated genes by comparing CD34-high samples to CD34-negative samples, with a fold change cutoff of 2.0 (n = 896 entities). Entities were exported to ToppCluster in order to identify enrichment in previously published microarray datasets (Coexpression Ontologies). A network consisting of genes and associated studies was generated through ToppCluster and Cytoscape.

## Chromatin immunoprecipitation

Chromatin immunoprecipitation was performed as previously described (*McCauley et al., 2014*). Briefly, NIH3T3 cells were seeded in 10 cm plates at 80% confluence and transiently transfected with a pCMV-driven mouse SMAD3 (Sino Biological Inc., Daxing, China) using X-treme Gene transfection reagent (Roche Applied Science, Indianapolis, IN, USA) for 24 hr, then treated with recombinant human TGF$\beta$1 (R & D Systems, Minneapolis, MN, USA, 2 ng/ml) for an additional 24 hr. Cells were cross-linked with 1% formaldehyde and subjected to ChIP using an antibody against SMAD3 (Abcam, Cambridge, MA, USA, RRID:AB_2192903) using a ChIP assay kit (Millipore, Billerica, MA, USA) according to manufacturer's instructions. After purification, DNA obtained from the ChIP assay was used as PCR templates to verify the interaction between DNA and protein, using primers designed to amplify distinct sites in the mouse *Elmo1* and *Dock2* promoters. Primers are described below. PCR products were then subjected to gel electrophoresis on a 3% agarose gel using a molecular weight marker to verify the size of migrating bands.

Primers used:

*Elmo1*

| | SBE F | gctttcttcagtccctcatagga |
|---|---|---|
| | SBE R | agcttccatttcagggaaactcc |
| | TIE F | gtgcaaccaggagaatttaaagca |
| | TIE R | tgagatgcccgaatccggag |

*Dock2*

| | | |
|---|---|---|
| | SBE4 F | caacttgtgctgtcagaaactgaa |
| | SBE4 R | tctcagggctaccatcacaatg |
| | SBE2/3 F | cattgtgatggtagccctgaga |
| | SBE2/3 R | ttctgtgccatgaacccaactg |
| | SBE1 F | gtcactaacagggttcagaagtca |
| | SBE1 R | atttggagaccaccctcatttgtc |
| | TIE F | gtcactcactgagtacaggttctt |
| | TIE R | tgacttctgaaccctgttagtgac |

## Plasmids and lentiviral infection

Using EcoRI and XhoI restriction enzymes, the full length Mus musculus *Tgfbr2* gene (1.7 kb) was isolated from a pcDNA3 expression vector (*Guasch et al., 2007*), sequenced and subcloned into the multi-cloning site of the pLVX-IRES-mCherry lentiviral vector (Clontech, Mountain View, CA) using the NEB Quick Ligation Kit (New England Biolabs, Ipswitch, MA), according to manufacturer's instructions. The resulting ligation was transformed into DH5α competent bacteria and selected on LB-amp plates overnight. DNA was extracted using a Maxi Prep DNA kit (Qiagen, Venlo, Limburg) according to manufacturer's instructions. Colonies were subjected to enzymatic digestion followed by sequencing to confirm the integration.

Vector control (pLVX-IRES-mCherry), rescue (pLVX-TGFβRII-IRES-mCherry) and hairpin resistant ELMO1 construct (pLVX-IRES-mCherry-ELMO1*) constructs were produced by the Cincinnati Children's Lentiviral Core and the lentivectors production facility/SFR BioSciences Gerland – Lyon Sud. 2 mL of viral supernatant were used for each 10 cm plate, after being washed in E Media with 0.05 mM Ca++ and concentrated by three centrifugations at 4000 rpm for 15 min at 4°C using a Vivaspin 20MWCO 30 kDa column (GE Healthcare, Pittsburgh, PA). Concentrated virus was combined with 8 µg/ml polybrene (Hexadimethrine bromide, Sigma. St. Louis, MO) and 3 ml E Media with 0.05 mM Ca++, applied to *Tgfbr2* cKO CD34⁺ SCC cells seeded at 60% confluency in 10 cm plates, and incubated at 37°C 5% CO2 for 24 hr. 24 hr after infection, plates were washed three times with sterile 1X PBS and given fresh E Media with 0.05 mM Ca++. 48 hr after infection, YFP+mCherry+ cells were selected by FACS and used directly for in vivo orthotopic transplantation or re-plated for in vitro experiments. To confirm the rescue, cells were treated with recombinant human TGFβ1 (R and D Systems, Minneapolis, MN, USA, 2 ng/ml) for 1 hr at 37°C before cells were trypsinized and processed for RNA extraction for qPCR or lysed for protein extraction and Western blot. Experiments were performed three times in triplicate and statistical significance was determined using paired two-tailed Student's t-test.

## shRNA knockdown

Two MISSION shRNA pLKO.1-puro bacterial constructs against the mouse *Elmo1* gene (TRCN0000112655, TRCN0000112656) and the SH02 control shRNA were purchased from Sigma Aldrich via the Cincinnati Children's Robotic Lenti-Library Core and lentivirus for SH02 control, TRCN0000112655 (ELMO1 shRNA construct #1) and TRCN0000112656 (ELMO1 shRNA construct #2) was produced by the Cincinnati Children's Lentiviral Core and the lentivectors production facility/SFR BioSciences Gerland – Lyon Sud. 2 mL of viral supernatant were used for each 10 cm plate, after being washed in E Media with 0.05 mM Ca++ and concentrated by three centrifugations at 4000 rpm for 15 min at 4°C using a Vivaspin 20MWCO 30 kDa column (GE Healthcare, Pittsburgh,

PA). Concentrated virus was combined with 8 µg/ml polybrene (Hexadimethrine bromide, Sigma) and 3 ml E Media with 0.05 mM Ca++, applied to *Tgfbr2* cKO CD34+ SCC cells seeded at 60% confluency in 10 cm plates, and incubated at 37°C 5% CO2 for 24 hr. 24 hr after infection, plates were washed three times with sterile 1X PBS and given fresh E Media with 0.05 mM Ca++. Beginning 48 hr after infection, 1 µg/ml puromycin was added with fresh E Media with 0.05 mM Ca++ every other day to select for infected puromycin-resistant clones.

## Mutagenesis assay

To create a construct that is not recognized by the *Elmo1* shRNA construct #2 (TRCN0000112656), we created three bases mutations in its target sequence without affecting ELMO1 function. The Geneart site-direct mutagenesis kit (Thermofisher Scientific) was used on pCMV-Sport6 expressing the full-length *Mus musculus Elmo1* gene (2,2 Kb) (GE Dharmacon), sequenced and subcloned into the multicloning site of the pLVX-IRES-mcherry lentiviral vector (clonetech, Mountain View, CA) using the NEB Quick Ligation Kit (New England Biolaps, Ipswitch, MA). Lentiviral vectors were produced by SFR Biosciences Gerland Lyon Sud.

Primers used to create the mutagenesis:

ELMO1 mut

F   AACTTGCTTTCTCCATCTTGTATGATTCAAATTGCCAACTGAACT

R   AGTTCAGTTGGCAATTTGAATCATACAAGATGGAGAAAGCAAGTT

## Quantification of lung metastasis by FACS

Lungs from tumor-bearing mice were dissociated into a single cell suspension according to a modified protocol based on one previously established in our lab (*McCauley and Guasch, 2013*). Briefly, lungs were inflated with a cocktail containing dispase (Sigma, St. Louis, MO), 20% collagenase (Sigma, St. Louis, MO) and Hank's buffered saline solution (Gibco, Waltham, MA) and dissociated in the same cocktail for 30 min while shaking at 37°C. Dissociated lung tissue was washed and filtered according to our tumor dissociation protocol (*McCauley and Guasch, 2013*), with the added treatment of red blood cell lysis buffer according to manufacturer's instructions (eBioscience, San Diego, CA). Whole lungs were sorted for YFP and analyzed using a FACS Aria II (BD Biosciences) and FACS-Diva software (BD Biosciences) in the Research Flow Cytometry Core at CCHMC. Percent YFP+ cells was calculated by dividing number of YFP+ cells by total cells after gating for appropriately sized single cells by forward scatter and side scatter. Experiments were performed twice in triplicate and statistical significance was determined using paired two-tailed Student's t-test.

## In vitro wound healing assay

The wound-healing assay was used to determine cell migration ability. $5 \times 10^4$ cells were plated in ibidi culture insert onto the 24-well plate (ibidi cat-80241) for 24 hr to reach 90–95% of confluence. A wound was created by removing inserts. Cells were washed with PBS twice to remove detached cells and incubated with medium E low $Ca^{2+}$ containing puromycin (1 µg/ml). The cells were observed under an inverted light microscope (Carl Zeiss) equipped with a CCD camera (Ropper) at X10 objective. Images were taken by MetaMorph software every 10 min for 10 hr. The wound widths of different area at each time points were measured with MetaMorph software. Data result from calculating the slope of linear trend curve of wound widths as a function of time and are representative of three experiments. Quantitative data are presented as the mean and significant difference was determined by two-tailed Student's t-test.

## RAC activity assay

Cell culture dishes at 80% confluence were washed with ice-cold PBS1X, lysed with 500 µl of lysis buffer (50 mM Tris, pH 7,2, 350 mM NaCl, 1% Triton X-100, 0.5% Na deoxycholate, 0.1% SDS, 10 mM $MgCl_2$, protease inhibitor cocktail complete tablets (Roche)) and centrifuged for 5 min at 13,000 RPM at 4°C. The supernatant was incubated with bacterially produced glutathione-S-transferase (GST)-PAK-CD fusion protein, containing the RAC and Cdc42-binding region from human PAK1B (*Sander et al., 1998*) bound to glutathione-coupled Sepharose beads at 4°C for 45 min. The beads and proteins bound to the fusion protein were washed three times with a wash buffer (50 mM Tris, pH7.2, 1% Triton X-100, 150 mM NaCl, 20 mM $MgCl_2$, protease inhibitor cocktail complete tablets

(Roche)) and eluted in Laemmli sample buffer (60 mM Tris pH 6,8, 2% SDS, 10% glycerine, 0,1% bromophenol blue) and then analysed for bound RAC1 molecules by western blot using a RAC1 antibody.

## Cell proliferation assay

Cells were stained with 10 μM Cell Proliferation dye eFluor 670 (Affymetrix eBioscience) according to the manufacturer's guidelines and analyzed by flow cytometry at 0 hr, 24 hr, 48 hr and 72 hr with a Fortessa instrument (3 lasers 405/488/630) (Becton Dickinson). DAPI staining was used and excluded to acquire $10^4$ cells in the live population. Three separate experiments have been done and the mean of the geometric mean for each sample has been calculated using Prism software (RRID:SCR_002798).

## Acknowledgements

We thank Dr. Kathryn Wikenheiser-Brokamp for pathological interpretation of tumor slides, Dr. Rebekah Karns for expertise in analyzing RNA-Seq data, Dr. Anil Jegga for expertise in analyzing promoter sequences, Dr. Michaël Sebbagh for help with the RAC assay, Pascal Finetti for database comparison and Jean-Mehdi Grangeon for the design of the *Figure 5—figure supplement 1*. Slides from human patients were generously provided by the University of Cincinnati. Flow cytometric data were acquired using equipment maintained by the Research Flow Cytometry Core in the Division of Rheumatology at Cincinnati Children's Hospital Medical Center (supported in part by NIH AR-47363, NIH DK78392 and NIH DK90971) and by the Flow Cytometry Core at the CRCM. Processing of the human slides was performed by the experimental histopathology (ICEP) core facilitiy at the Institut Paoli Calmettes. We thank Daniel Isnardon in the microscopic core facility at the CRCM and Gisèle Froment, Didier Nègre and Caroline Costa from the lentivectors production facility/SFR BioSciences Gerland – Lyon Sud (UMS3444/US8). This work was supported by grants from the V Foundation, the Sidney Kimmel Foundation and in part from the Fondation ARC pour la recherche sur le cancer (GG).

## Additional information

### Funding

| Funder | Grant reference number | Author |
| --- | --- | --- |
| Fondation ARC pour la Recherche sur le Cancer | PJA20141201606 | Géraldine Guasch |
| V Foundation for Cancer Research | | Géraldine Guasch |
| Sidney Kimmel Foundation for Cancer Research | | Géraldine Guasch |

The funders had no role in study design, data collection and interpretation, or the decision to submit the work for publication.

### Author contributions

HAM, Conceptualization, Validation, Writing—original draft, Writing—review and editing; VC, Conceptualization, Validation; DB, Formal analysis, Validation, Writing—review and editing; GG, Conceptualization, Formal analysis, Supervision, Funding acquisition, Validation, Investigation, Methodology, Writing—original draft, Writing—review and editing

### Author ORCIDs

Heather A McCauley, http://orcid.org/0000-0002-6834-4091
Géraldine Guasch, http://orcid.org/0000-0001-7362-9318

### Ethics

Animal experimentation: All experiments were approved by the Cincinnati Children's Hospital Research Foundation Institutional Animal Care and Use Committee (protocol number 1D10087) and

in agreement with European and national regulation (protocol number 4572) and carried out using standard procedures

## Additional files

### Supplementary files

• Supplementary file 1. *Tgfbr2* cKO CD34+ SCC signature. Annotated genes for mRNAs upregulated and downregulated more than two fold with an FDR < 0.05 in FACS-purified CD34+ epithelial cancer populations in comparison with mRNA from CD34− epithelial cancer populations purified with the same surface markers. Each column represents the average of two data sets for CD34+ and CD34− populations obtained from independent SCCs. Gene names are listed from the most highly upregulated to the least. Averaged expression values and averaged fold change (FC) are listed.

### Major datasets

The following previously published datasets were used:

| Author(s) | Year | Dataset title | Dataset URL | Database, license, and accessibility information |
|---|---|---|---|---|
| Schober M, Fuchs E | 2011 | Expression data from squamous cell carcinoma stem cells, epidermal progenitor cells and hair follicle bulge stem cells | http://www.ncbi.nlm.nih.gov/geo/query/acc.cgi?acc=GSE29328 | Publicly available at NCBI Gene Expression Omnibus (accession no: GSE29328) |
| Beck B1, Driessens G, Goossens S, Youssef KK, Kuchnio A, Caauwe A, Sotiropoulou PA, Loges S, Lapouge G, Candi A, Mascre G, Drogat B, Dekoninck S, Haigh JJ, Carmeliet P, Blanpain C. | 2011 | Expression data from sorted epithelial CD34+ expressing cells from DMBA/TPA induced skin tumors | https://www.ncbi.nlm.nih.gov/geo/query/acc.cgi?acc=GSE31465 | Publicly available at NCBI Gene Expression Omnibus (accession no: GSE31465) |

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
