## [Decision Letter]

Thank you for submitting your article "De-repression of ELMO1 in cancer stem cells drives progression of TGFβ-deficient carcinoma from transition zones" for consideration by *eLife*. Your article has been reviewed by two peer reviewers, and the evaluation has been overseen by a Reviewing Editor and Fiona Watt as the Senior Editor. The following individual involved in review of your submission has agreed to reveal his identity: Xiaoyang Wu (Reviewer #2).

The reviewers have discussed the reviews with one another and the Reviewing Editor has drafted this decision to help you prepare a revised submission

Summary:

In this manuscript, the authors analyze the molecular mechanisms that control the high metastatic potential of cells in transitional epithelia. They show that a CD34 high potential cancer stem cell population exists in the transition zone SCC and that TGFβ receptor II loss activates expression of ELMO1, a GEF for Rho family small GTPase protein, identifying a pathway critically for the invasiveness and metastasis of transitional zone SCC. Overall, the reviewers were intrigued by the hypothesis and impressed by the data that identifies a potential cancer stem/initiating cell population and defines the molecular mechanisms that initiate metastatic behavior of these cells.

Essential revisions:

1) ELMO1 is a GEF for Rac small GTPase. Instead of transcriptionally upregulating Rac, ELMO1 activates Rac by GTP loading. However, this has not been tested in the manuscript, whether Rac activity is indeed upregulated in cancer stem cells of transitional zone SCC.

2) In reference to Figure 5—figure supplement 1, the authors should compare the gene expression profiles with the previously described transcriptional profiles in Schober et al. 2011 since the data use the same genetic drivers (Tgfbr2 KO). Furthermore, comparing these data with the other transcriptional profiles of CSC from skin cancers including Lapouge et al. EmboJ 2012 and Boumahdi et al. Nature 2014, as they presented profiling of CD34+ and CD34- as well as *Sox2*^+^CD34+ CSCs in these papers.

3) The authors should add discussion points that compare the transitional CSCs with the previous papers that functionally characterize CSC in the skin.

---

## [Author Response]

*Essential revisions:*

*1) ELMO1 is a GEF for Rac small GTPase. Instead of transcriptionally upregulating Rac, ELMO1 activates Rac by GTP loading. However, this has not been tested in the manuscript, whether Rac activity is indeed upregulated in cancer stem cells of transitional zone SCC.*

We have performed a Rac activity assay (new Figure 4) and show that cancer stem cells of transitional zone SCC have a strong Rac1 activity. We have added a new Rac activity assay in the Materials and methods section.

*2) In reference to Figure 5—figure supplement 1, the authors should compare the gene expression profiles with the previously described transcriptional profiles in Schober et al. 2011 since the data use the same genetic drivers (Tgfbr2 KO). Furthermore, comparing these data with the other transcriptional profiles of CSC from skin cancers including Lapouge et al. EmboJ 2012 and Boumahdi et al. Nature 2014, as they presented profiling of CD34+ and CD34- as well as Sox2Sox2^+^CD34+ CSCs in these papers.*

As suggested by the reviewer we have now compared our CD34+ and CD34- anorectal TZ SCC TGFb-deficient signature with all databases in Schober et al., 2011 that are in the same genetic drivers TGFbr2 KO with and without loss of FAK. We have also compared our CD34+ and CD34- anorectal TZ SCC TGFb-deficient signature with other transcriptional profiles of CSC from skin cancers in a TGFb-intact background (DMBA/TPA treated) in a WT background, VEGF gain of function (from Beck et al., 2011) and *Sox2* cKO background (Boumahdi et al., 2014). There were only 38 genes available in the study using *Sox2* deletion and unsurprisingly we did not find any genes in common with a p value<0.05, so we did not include a Venn diagram with these data. Moreover, there are no data from skin SCC CD34+ open to the public in Lapouge et al., Embo J 2011.

Subsection “Upregulation of RAC signaling is unique to the anorectal *Tgfbr2* cKO SCC and not found in *Hras*-induced skin SCC” in the Results part has been modified accordingly with the new Venn Diagrams in the new Figure 5—figure supplement 1.

*3) The authors should add discussion points that compare the transitional CSCs with the previous papers that functionally characterize CSC in the skin.*

We have added new Discussion points that compare our transition zone CSC signature with CSC from skin SCC from various genetic backgrounds (WT, FAK KO, Gain of function of VEGF, Tgfbr2-deficient, and Tgfbr2/FAK double KO).